# Behavioral and Oxidative Stress Changes in Mice Subjected to Combinations of Multiple Stressors Relevant to Irritable Bowel Syndrome

**DOI:** 10.3390/brainsci10110865

**Published:** 2020-11-17

**Authors:** Roxana Oana Cojocariu, Ioana Miruna Balmus, Radu Lefter, Daniela Carmen Ababei, Alin Ciobica, Luminita Hritcu, Fatimazahra Kamal, Bogdan Doroftei

**Affiliations:** 1Department of Biology, Faculty of Biology, Alexandru Ioan Cuza University, 11th Carol I Avenue, 700506 Iasi, Romania; roxana_20_2006@yahoo.com; 2Department of Interdisciplinary Research in Science, “Alexandru Ioan Cuza” University of Iasi, Carol I Avenue, no. 11, 700506 Iasi, Romania; balmus.ioanamiruna@yahoo.com; 3Center of Biomedical Research, Romanian Academy, 8th Carol I Avenue, 700506 Iasi, Romania; radu_lefter@yahoo.com; 4“Grigore T. Popa” University of Medicine and Pharmacy, 16th Universitatii Street, 700115 Iasi, Romania; dana.ababei@gmail.com; 5Faculty of Veterinary Medicine, University of Agricultural Sciencies and Veterinary Medicine “Ion Ionescu de la Brad” of Iasi, 3rd Mihail Sadoveanu, 700490 Iasi, Romania; 6Faculty of Sciences and Technology Settat, University of Hasan I, B.P. 539, 26000 Settat, Morocco; fatimzahra.kamal@gmail.com; 7Department of Obstetrics and Gynecology, Faculty of Medicine, “Gr. T. Popa” University of Medicine and Pharmacy, 16th University Street, 700115 Iasi, Romania; bogdandoroftei@gmail.com; 8Origyn Fertility Center, Human Reproduction, Palace Street, No. 3C, 700032 Iasi, Romania

**Keywords:** irritable bowel syndrome, neonatal maternal separation, chronic unpredictable mild stress, restraint stress, behavioral tasks, oxidative stress, mice

## Abstract

*Background and Objectives*: Irritable bowel syndrome (IBS) is a well-known functional gastrointestinal (GI) disorder exhibiting a wide range of symptoms due to individual variability and multifactorial etiology. Stress exposure is a major risk factor for the development of IBS. Here, we investigate the differential effects of psychological stress exposures on behavior and oxidative status in mice by using increasingly complex combinations of etiologic IBS-relevant stressors (maternal separation and chronic unpredictable mild stress combinations). *Materials and Methods*: Mice were subjected to three different combinations of psychological stress factors and subsequent behavioral cognitive and affective parameters and oxidative status markers (superoxide dismutase and glutathione peroxidase antioxidant activity and malondialdehyde level) in the brain and bowel tissues of the animals were analyzed. *Results*: GI transit modifications reflected by decreased fecal output, cognitive and affective behavioral deficits were observed in all stress exposed groups, but were more evident for the more complex combinations of stressors. Behavioral deficits were accompanied by mild oxidative stress occurring in the bowel and to a greater extent in brain tissue. *Conclusions*: The presented data depict the effect of various associations in mimicking IBS symptoms and comorbidities and suggest that an all-inclusive combination of early and adult-life psychological stressors is more effective in IBS symptoms modulation. Oxidative stress in both brain and bowel, suggestive for brain-gut molecular connectivity, may play an important role in IBS mechanistic.

## 1. Introduction 

Irritable bowel syndrome (IBS) is a well-known functional gastrointestinal (GI) disorder exhibiting a wide range of symptoms due to individual variability and multifactorial etiology [1]. The most common symptoms include bowel habits changes accompanied by abdominal pain in the absence of detectable tissular changes [2]. Psychological stress is an incriminating factor in the occurrence of IBS symptoms [3] and is correlated with dysregulation of hypothalamic–pituitary–adrenal axis (HPA) response which in turn can induce visceral hyperalgesia, altered intestinal transit and colonic motility in both humans or animals [4]. The brain-gut axis impairment and the stress-related physiological modulation of the HPA axis have been previously described as relevant to IBS development [5,6]. In addition, the brain-gut axis impairments are common in many psychiatric disorders, such as the mood disorders, anxiety, depression [7,8] and autism [9]. For a better understanding of the mechanisms underlying IBS, psychosocial stressing have been used to develop several animal models that reproduce parts of the IBS-like symptoms [10]. Thus, daily contention in plastic tubes for a limited period or exposure to multiple variable stressors are used to induce visceral hypersensitivity and irregular intestinal transit [11,12], which are accompanied by altered affective behaviors, such as anxiety and depression [13]. Neonatal maternal separation of rodents, another well-known method to replicate IBS pathophysiology, is described to also alter HPA axis-related emotional responses in later life [14]. 

The characteristics of the various psychological stressors must be carefully considered when applied in IBS animal models, as they may result in some disadvantages. Unlike the chronic and milder combinations, acute single stressors, such as restraint stress, may fail in altering some molecular pathways that may be responsible for the pathogenesis of IBS [15], and if prolonged, the intense stimulation can cause somatic damages [16]. At the same time, exposure of animals to chronic stress may lead to habituation to the repetitive stimuli [16]. Developing combined animal models using a variety of unpredictable stressors should reduce habituating patterns and be more representative for the multiple pathogeny of IBS [16]. In a previous report on stressed-induced depressive behavior in a chronic unpredictable mild stress rat model [17], our group discussed the relevance of chronic stress, which could be correlated to the multiple impairments caused by environmental and psychological stressors. However, a more effective approach when modeling IBS in animals by psychosocial factors may ensue when considering the diathesis-stress model that explains psychiatric disorders as complex interactions between genetic predisposition and stressful events [18]. Vulnerability of the nervous system during early life to negative events such as maternal separation may only augment and lengthen stress responses in later development, as [19] demonstrated in socially-isolated juvenile rats. 

The fact that no specific IBS biomarker has been described to date has led to an increased interest for oxidative status. Both signaling impairments and oxidative balance changes have been reported in several neurological and GI disorders [20,21] and recent studies suggested that oxygen reactive species (ROS) and the antioxidant system imbalance could be implicated in IBS development [22,23]. Significant variations of oxidative stress enzymes leading to a decrease of antioxidant capacity were observed in IBS patients [23,24]. Various IBS rat models induced by either psychological stress or chemical stimulation were also reported to exhibit increases in lipid peroxidation and decreased superoxide dismutase (SOD) and glutathione peroxidase (GPx) activities in bowel tissue [11,25]. 

In the current study we aimed to investigate the effect of stress exposure on behavior, intestinal transit and oxidative status in mice by using combinations of previously validated IBS stress-based paradigms. Our hypothesis was that the intensity of cumulative stress comprising of early and adult life stressors would alter intestinal motility, affective and cognitive states in mice more than either of the separate stressors. We also assumed that a disturbed antioxidative balance in brain and in colon as a result of psychological stress would accompany and correlate with behavioral changes, which might help to explain the biochemical mechanisms underlying IBS.

## 2. Materials and Methods

### 2.1. Animal Housing and Habituation

Male Swiss mice at an initial body mass of 30–40 g were habituated in constant environmental conditions (20 °C, 55–60% humidity, natural light-dark cycle and free access to water and food) in polyacrylic cages (5 animals/cage) containing woodchip bedding. Romanian and the European laws on animal use in biomedical research were considered in animal care and experimental procedures. This study was approved by the local committee (no. USAMV Iasi 385/04.04.2019) and efforts were made to reduce the number of animals and their suffering.

### 2.2. Experimental Design

Forty mouse male pups were selected and four groups (*n* = 10) were created for the current experiment. Three groups were subsequently exposed to different stress combinations—as described below—and established as IBS animal models. The remaining group served as the control group and was subjected to identical environmental conditions in the absence of any studied stress factors. 

Two of the three IBS groups (*n* = 20), were subjected to neonatal maternal separation (MS) for 3 h/day between postnatal days (PD) 1 and 14. The third IBS group and Control group were left unhandled. Beginning with postnatal day 90, the third IBS group and one of the previously MS groups were subjected to 7 days (PD 90–96) of combined multifactorial stress-exposure consisting of (a) unpredictable mild stressors and (b) a repetitive stress factor. Along the 7 days of stress-exposure the (a)-type stressors were applied during the morning—with the exception of the food/water deprivation stressors that were persistent throughout a 24 h period (Table 1). In the second part of the day mice were subjected for 1 h/day to the (b) type stressor represented by water avoidance stress paradigm. The sequence of unpredictable mild stressors included: (1) restraint stress [16], 30 min/day, (2) exposure to sound predator (birds of prey cries lasting 10 min at ambient level), (3) 24 h water deprivation, (4) injection simulation, (5) tilt cages backwards at 45 degrees during 1 to 4 h, (6) 1 min tail pinch at 1 cm from the end of the tail, (7) 24 h food deprivation. The one-hour water avoidance stress (WAS) procedure consisted in placing each mouse on a small platform (2.5 cm diameter) in the middle of a small plastic basin filled with warm water (22 °C) at the height level of the platform. Control group mice were placed on the same platform but in a waterless container for 1 h. 

Following stress exposure, all the animals were subjected to behavioral assessment during PD 101—PD 109 in the following order: Y Maze Test in PD 101, Elevated plus Maze in PD103 and Forced Swim Test in PD106—PD109. The biological samples were collected in PD 111 (Figure 1). The woodchip bedding from the cages was changed periodically. After the last stress session all the fecal pellets were collected, counted, and evaluated in terms of consistency 24 h. The evaluation of fecal output was conducted for each cage of animals not individually. Animals were divided into four groups depending on the type of stressor applied: (1) control, (2) neonatal maternal separation + multifactorial stress exposed group (MS+MF), (3) multifactorial stress exposed group (MF), (4) maternal separation (MS) group. Early-age stress (maternal separation) reflecting a certain genetic vulnerability that would add to the burden of the later stressful events constituted one of the stressors (MS), often cited as risk factor in IBS [26]; multiple chronic stressors associated with restraint stress and water avoidance stress in adult stage made up the second combination suggestive for a highly stressful adult environment (MF+WAS); the third combination was the most complex by merging all the above stressors (MS+MF), in order to replicate a diathesis-stress model based on genetic predisposition and highly stressful events. 

### 2.3. Behavioral Testing

Following animal model development, the animals were subjected to behavioral tests in the following order: Y maze test, elevated-plus maze and forced swim test.

#### 2.3.1. Y Maze Test 

Y maze test was used for short-term memory evaluation by assessing the exploratory behavior of the three arms of the Y-shaped apparatus, according to the protocol described by Kokkinidis group [27]. The maze used in the present study consisted of three arms (40 cm length, 8 cm width, and 15 cm height, attached at 120 degrees angles) and an equilateral triangular central area. Each mouse was placed at the end of one arm and allowed to freely explore the maze for 8 min. For evaluating short-term memory we have assessed the spontaneous alternation indicator, defined by the entries into all three arms on consecutive choices. For further statistical analyses the spontaneous alternation percentage (%) was calculated as the ratio of spontaneous alternation entries per total entries. 

#### 2.3.2. Elevated Plus Maze

Elevated maze test (EPM), a cross-like four arms apparatus elevated 50 cm off the ground with two arms enclosed by walls of 30 cm high and the other two exposed was used to assess anxious behaviors. Each animal was placed at the juncture of the open and closed arms and was let to freely explore the maze for 5 min. During this period the entries and time spent in each arm and grooming bouts were recorded as indicators for anxiety recorded during a 5-min test according to the protocol described by Pellow group [28]. 

#### 2.3.3. Forced Swim Test 

Behavioral despair was assessed using an adapted variant of Porsolt’s forced swim test (FST) [29] for mice. The protocol consists in maintaining the individuals in a transparent glass cylinder (30 cm in diameter, height 59 cm) filled with water (15 cm, 26 °C) while the swimming patterns of the escape behavior are assessed. The animals were exposed to the experimental conditions for 6 min, comprising of the first two minutes for acclimatization and the last 4 min for measuring the a series of behavioral parameters that serve as indicators of depressive state: swimming, immobility (floating) and struggling behavior.

### 2.4. Tissue Collection and Preparation

Biological samples were obtained in the following day after behavioral evaluation in total anaesthesia conditions (ketamine 100 mg/kg, xylazine 10 mg/kg). Brain and colon were collected from the animals and the colon was emptied of all content and washed twice with physiological saline. Following this, 0.2 g of tissues were subjected to extract preparation using tissue extraction buffer (0.328 g TRIS, 1.304 g KCl_2_ and distilled water to 200 mL volume, pH = 7.4) and subjected to biochemical assessment.

### 2.5. Biochemical Determinations 

SOD enzymatic activity was determined using a spectrophotometric SOD Assay Kit (Sigma, Darmstad, Germany) according to the manufacturer’s instructions. The indirect measurement of SOD activity was obtained based on the water soluble tetrazolium (WST) salt reaction with superoxide anion producing a water-soluble formazan dye.

GPx enzyme activity was assessed using the GPx Cellular Activity Assay Kit CGP-1 (Sigma). The indirect determination method of the GPx activity is based on the nicotinamide adenine dinucleotide phosphate (NAPDH) concentration decrease in the reaction media during which NADPH is oxidized to NADP+. 

Lipid peroxidation as reflected by malondialdehyde (MDA) levels was assessed using thiobarbituric acid-reactive substances (TBARs) determination method [30]. Trichloroacetic acid (50%, 0.25 mL), thiobarbituric acid (0.73%, 0.255 mL) and tissue extracts (0.05 mL) were mixed and vortexed. Afterwards, a 20 min incubation at 100 °C (boiling water bath), and a 10 min centrifugation (3000 rpm) were performed. The supernatants were exposed to a 532 nm spectrometry system and the absorbance was read against MDA standard curve (the results were expressed as mmol MDA/mL tissue extract). 

### 2.6. Statistical Analysis 

The numerical data obtained by behavioral and biochemical evaluation was statistically analyzed using one-way analysis of variance (ANOVA) using Minitab 17 (Pennsylvania University, US, 2017). The results are expressed as means ± SEM and were regarded statistically significant at *p* < 0.05. Statistical correlations were expressed as Pearson’s linearity coefficient while *p* < 0.05. Post-hoc analysis included Bonferroni corrected student *t*-test. 

## 3. Results 

### 3.1. Behavioral Parameters Evaluation

#### 3.1.1. The Effect of Various Combinations of Stress Factors on Gastrointestinal Tract Habits 

Regarding the animals’ bowel habits changes we observed that the exposure to different stress factors combinations can induce significant increases in bowel transit time. Thus, slowed defecation observed by fecal pellet count/24 h after stress exposure appeared in all three cases stressors combinations as compared to control group (Figure 2). However, the differences vs. control were more clear for the combination of early-life maternal separation stress and adult-life multifactorial stress for MS+MF vs. control (F (1, 18) = 24.13, *p* = 0.001). 

#### 3.1.2. The Effects of Various Combinations of Stress Factors in the Y-Maze Test

The behavioral analysis of the short-term memory performance showed no significant overall differences between the experimental groups (F (3, 36) = 2.28, *p* = 0.09) in terms of spontaneous alternation (%). However, post-hoc analysis showed significant decreases of spontaneous alternation in the MF group (F (1, 18) = 4.42, *p* = 0.04; t(18) = 2.11, *p* = 0.02) and MS+MF group (F (1, 18) = 5.65, *p* = 0.02; t(18) = 2.03, *p* = 0.03), as compared to control group (Figure 3). 

#### 3.1.3. The Effects of Various Combinations of Stress Factors on the Parameters Evaluated in Elevated Plus Maze

Regarding the number of open arms entries, a parameter for which lower values are consistent with anxiogenic–like behavior since it is based on the natural aversion of rodents for open, visible decreases were observed for all stress exposed groups as compared to control group, which were statistically significant in the MS+MF group (F (1, 18) = 6.18, *p* = 0.023; t(9) = 2.75, *p* = 0.01) and almost significant in the MS group (F (1, 18) = 3.92, *p* = 0.063; t(16) = 1.97, *p* = 0.03) (Figure 4a). The time spent in open arms (Figure 4b)—one of the most suggestive indicators for anxiolytic effects—was visually decreased for all stress-exposed groups when compared to control group, but significantly decreased only for MF (F (1, 18) = 4.52, *p* = 0.047; t(10) = 2.12, *p* = 0.02) and MS+MF groups (F (1, 18) = 4.42, *p* = 0.049; t(18) = 2.10, *p* = 0.03) vs. control. Interestingly, when we analyzed the number of entries in closed arms (Figure 4c), a locomotor oriented parameter, we found a significant decrease in the exploration of the maze’s closed arms in the MS group (F (1, 18) = 5.44, *p* = 0.037; t(16) = 2.33, *p* = 0.01) vs. control, whereas the other two stress-exposed groups showed less significant decreases. Moreover, the grooming bouts (Figure 4d), another anxiety indicator, showed no significant variations between groups, except for a moderate increase in the same MS group (F (1, 18) = 4.64, *p* = 0.044; t(12) = −2.15, *p* = 0.01) vs. control.

#### 3.1.4. Effects of Various Combinations of Stress Factors on the Parameters Evaluated in Forced Swim Test 

In the forced swim test, statistical analysis of the swimming time showed significantly low mobility for MS group (F (1, 18) = 6.69, *p* = 0.018; t(17) = 2.58, *p* = 0.009) and almost significantly low mobility for the other two stressed groups (Figure 5a). Post-hoc analysis showed significantly overall differences between groups for the floating time (F (3, 36) = 2.874, *p* = 0.04) and, reversely, increased floating time in all three stress-exposed groups, significantly increased in the MF (F (1, 18) = 3.64, *p* = 0.048; t(18) = −2.10, *p* = 0.02) and respectively MS+MF groups (F (1, 18) = 6.614, *p* = 0.019; t(14) = −2.57, *p* = 0.01) vs. control (Figure 5b). Moreover, we observed a significant group difference in terms of struggling duration (F (3, 36) = 5.433, *p* = 0.03), and a significant decrease in MS+MF vs. control group (F (1, 18) = 7.154, *p* = 0.015; t(13) = 2.67, *p* = 0.009) and also vs. MS group (F (1, 18) = 4.21, *p* = 0.05; t(10) = 3.62, *p* = 0.02) (Figure 5c). Interestingly, we observed an almost significant increase in struggling, which may be suggestive of anxious behavior, in the MS group, as compared to control (F (1, 18) = 3.928, *p* = 0.06; t(13) = −1.98, *p* = 0.03). 

### 3.2. Brain Tissue Biochemical Parameters Evaluation 

Regarding the brain oxidative stress status, post-hoc analysis showed no significant variations between groups in the antioxidant activity of SOD (Figure 6a), although visually slightly lower SOD values were obtained in the MS+MF group (F (1, 18) = 4.27, *p* = 0.53; t(18) = 2.06, *p* = 0.02) vs. control and slightly higher in the MF group. However, the GPx levels were generally decreased in the stress-exposed groups, and significant decreases were detected in the MF and MS+MF groups compared to the control group (F (1, 18) = 15.33, *p* = 0.002; t(12) = 3.916194657, *p* = 0.002) and (F (1, 18) = 8.44, *p* = 0.009; t(17) = 2.90, *p* = 0.04) respectively (Figure 6b). In what concerns brain MDA levels, significant overall differences were observed (F (3, 36) = 3.21, *p* = 0.03) suggesting the effect of stress exposure on this oxidative stress parameter. We found significantly increased MDA levels in the MF group (F (1, 18) = 8.604, *p* = 0.008; t(15) = −2.93, *p* = 0.005) vs. control and MS+MF (F (1, 18) = 7.516, *p* = 0.013; t(14) = −2.74, *p* = 0.007) vs. control, but not in the MS group (Figure 6c). 

### 3.3. Bowel Tissue Biochemical Parameters Evaluation 

When evaluating the oxidative stress markers in bowel tissues, no significant differences in SOD specific activity were detected between groups (Figure 7a). Similarly, GPx enzymatic activity in the stressed groups did not vary significantly relative to control, except for a significantly decrease in the MS group (F (1, 18) = 7.77, *p* = 0.012; t(18) = 2.788445607, *p* = 0.006) (Figure 7b). Regarding the colon MDA levels, a significant elevation was observed in the MS+MF group (F (1, 18) = 7.241, *p* = 0.0149; t(15) = −2.69104995, *p* = 0.008), whereas the other two stress-exposed groups had no significant increases in comparison with the controls (Figure 7c).

### 3.4. Behavioral and Biochemical Parameters Correlations 

Correlation analysis (considering Pearson’s linearity coefficient) revealed several significant correlations between behavioral and biochemical parameters. In this way, we found moderate negative or positive correlation between depression and anxiety indicators, such as floating duration in FST versus frequency of open arms entries in EPM (*r* = −0.705, while *p* < 0.0001) (Figure 8b), mobility duration in FST and frequency of open arms entries (*r* = 0.649, *p* < 0.0001) (Figure 8a), mobility duration and open arms time (*r* = 0.602, *p* < 0.0001) (Figure 8c). Correlations between the spontaneous alternation memory indicator and other indicators were mild (*r* = −0. 437 vs. open arms entries; *r* = 0.454 vs. swimming duration) or weak. 

When we analyzed Pearson’s linearity correlations between behavioral and oxidative stress parameters (Figure 9), we found a series of moderate correlations for anxiety and depression indicators vs. brain tissue oxidative biomarkers, such as: moderate positive correlation between floating duration and MDA brain levels (*r* = 0.731, *p* < 0.001) (Figure 9a), moderate negative correlation between swimming and MDA brain levels (*r* = −0.653, *p* < 0.001), moderate negative correlation between floating and GPx activity (*r* = −0.694, *p* < 0.001) (Figure 9c), moderate negative correlation between open arms time and MDA brain levels (*r* = −0.615, *p* < 0.0001) (Figure 9b), moderate positive correlation between open arms entries and GPx activity (*r* = −0.582, *p* < 0.001) (Figure 9e), moderate positive correlation between the spontaneous alternation and brain SOD activity (*r* = 0.685, *p* < 0.01) (Figure 9d). In terms of correlations between behavioral and oxidative stress parameters evaluated from colon, we found a moderate degree of positive correlation between floating duration and bowel MDA levels (*r* = 0.558, *p* < 0.001) (Figure 9f). 

Furthermore, we analyzed Pearson’s correlational significance between oxidative stress parameters evaluated from brain and bowel tissues. Thus, we found a moderate negative correlation between GPx activity in brain and MDA levels in brain (*r* = −0.725, while *p* = 0.001) (Figure 10a), and similarly a mild negative correlation between GPx activity in bowel and MDA levels in bowel tissue (*r* = −0.505, *p* = 0.001) (Figure 10b). Also, we found moderate negative correlations between brain GPx activity versus MDA levels in bowel (*r* = −0.524, while *p* = 0.001) (Figure 10c), weak positive correlations between brain GPx and bowel GPx activity (*r* = −0.458, while *p* = 0.002) (Figure 10d) and weak positive correlations between brain MDA and bowel MDA levels (*r* = −0.479, while *p* = 0.001) (Figure 10e). 

## 4. Discussion 

The present study was designed to evaluate the effectiveness of three different combinations of stress factors relevant for mimicking IBS typical manifestations in rodents. Our results showed that all stress combinations chronically slowed intestinal transit and led to depressive and anxiety-like behaviors mimicking the IBS comorbidities [31]. On the whole, a cumulative stress comprising of early and later life stressors was the more effective in modulating gastroenterological features accompanied by depressive and anxious-like behavioral changes; however, the multifactorial stress (MF) exposure was seemingly a more determinant factor in modulating behavioral symptoms and oxidative stress, as often the supplementary stress load in the MS+MF combination did not differ significantly in symptoms exhibition. Lipid peroxidation increases were observed in brain and bowel tissues, suggesting that mild oxidative stress occur in intestinal cells as a result to stress exposure, but to a greater extent in brain tissue. Furthermore, these biochemical changes, in particular at brain level, can be correlated to a certain degree with behavioral symptoms.

With regard to the GI symptoms occurrence, our initial assumption was confirmed, that the higher the intensity of the stress factors combination, the more efficient would its effect be in altering the GI habits. A combination between restraint stress and maternal separation could lead to significantly slowed intestinal transit, as compared to control group, as demonstrated by our data regarding the fecal pellets count per 24 h. The modifications in the GI transit we observed in mice following the chronic stress-exposure are suggestive of slow-transit constipation, but, interestingly, during and shortly after the restraint stress routine mice presented acute diarrhea that did not persisted more than 30 min. These results are similar to those of [32] who reported watery diarrhea in rats after restraint stress. A large body of literature data showed that in rodents stress provokes stimulation of large intestinal activity and increased fecal excretion [33,34,35], however, it remains elusive how different forms and combinations of stress modulate colonic motility in animals. German et al. demonstrated the tendency towards diarrhea and constipation in shelters contained cats and suggested stress due to relocation into the shelter as a possible cause of fecal retention [36]. Also, [37] actually reported that the initially accelerated colonic transit caused by restraint stress in rats is slowed down gradually under chronic exposure due to adaptation. Several other studies in mice report similarly decreased fecal pellet output following early-life maternal separation stress [38] or chronic psychosocial stress [39,40] and suggest the timepoint as an important factor to be considered when measuring colonic motor function, due to fast adaptations in the colon in response to chronic stress [40]. 

The presented data also delineate the deleterious effects of stress exposure on affective states, which correspond to the abnormalities observed in GI functioning. Zhang group [41] recently reported the correlation between slowed intestinal transit (constipation) and depressive behaviors in a chronic stress-induced depression rat model. The idea of a correlation between the affective status and GI response has been highlighted since 1968 when Lieblich and Guttman showed that defecation intensity is strongly associated to specific emotional context response [42]. In the present study, all three stress factors-exposed groups were significantly vulnerable to depressive-like states, as suggested by the decreased swimming in the MS group and increased floating durations in the MF and MS+MF groups. The non-uniformity exhibited by the stressed groups when considering struggling is more difficult to interpret, given also the ambiguous significance of this parameter. In the literature, swimming is considered sensitive to serotonergic compounds like selective serotonin (5-HT) reuptake inhibitors and 5-HT receptor antagonists, while struggling is sensitive to tricyclic antidepressants and drugs with effects on catecholaminergic transmission [43,44]. If we regard struggling as an escape-directed behavior driven by anxiety state [45], we could presume that the MS stress factor nature could predispose to anxious behavior, whereas an increased stressful load, as in the case of the MS+MF group, would overcome anxiety threshold and shift it to depressive behavior, when struggling significantly decreases and becomes relevant for installation of a depressive state in which individuals no longer try to escape. 

Regarding the anxiety assessment in the elevated plus maze test, we observed some significant changes in the exhibition of anxiogenic behaviors. Based on the significant lessened tendency of entering/staying in the open arms of the maze, anxiety-like responses were more intense in the MF and MS+MF group than in the MS group. This role of chronic multifactorial stress and restraint stress in producing anxiety-like behavior, has been reported previously in rodents following restraint stress [46] or chronic stress exposure [47]. On another note, MS group exhibited a significant decrease in the closed arm entries during the EPM task. While closed arm entries may serve as a measure of spontaneous motor activity [48], in this case they may actually be relevant for the installation of anxiety-states, if we consider the increased grooming frequency measured for this group. According to a study of Yoon et al. [49] on chronic non-social stress exposure in mice, the depression and anxiety-related circuits modified by stress can be dissociated in the mouse brain and different stress types address different brain circuits, according to their social/non-social profile. In this light, we may suggest that the stress combinations in our study modulated affective states with largely similar outcome, but, due to being based on stressors with distinct social relevance (the social MS stressor vs. the non-social MF stressors), they activated preferentially different brain circuits, hence the variation of phenotypic manifestations. 

Regarding the short-term memory, our results indicated that stress exposure generally exerts an altering effect on cognitive process; however, the most significant effects were observed in the case of multifactorial stress and MF combined with MS. Short-term memory impairment was previously obtained by applying restraint stress [50] and by multifactorial unpredictable stress [51,52] in animal models, and may be the consequence of elevated activity of HPA axis and increase in stress hormones levels [53]. We have found less significant deficits in short-term memory during the Y maze task in the case of maternal separation exposure. The disruption of the mother–infant relationship exhibits life-long influence on the behavioral and endocrine responses to stress [54], however, the effect of maternal separation on short-term memory performance is controversial, and decreased cortisol levels were reported in maternal separated juvenile squirrel monkeys [55]. 

Mild inflammatory response and low intensity oxidative stress were described in IBS pathology, as reflected by increased colon mast cell density [56], low activity of plasma antioxidant enzymes and significantly higher MDA and NO concentrations [24,57]. We hypothesized that the psychological stressors we used would produce alterations in the brain and colon oxidative status that would be also reflected by behavioral modifications. In the literature, the association between oxidative stress and systemic inflammation and chronic unpredictable mild stress or chronic restraint stress exposure was previously documented in rodents [58,59,60]. We have found significantly increased lipid peroxidation processes in the brain in the MF and MS+MF groups, and less accentuated lipid peroxidation in the bowel tissues, with significant increase only in the MS+MF group. The antioxidant enzymes SOD did not register significant variations for these groups, but GPx activity at the brain level was significantly decreased in both groups and also statistically correlated with the increased MDA levels. Brain susceptibility to oxidative injury due to its structural particularities and increased oxygen consumption, previously documented [58], may explain the association between chronic stress exposure and oxidative stress more tangible in the brain. No significant variations in oxidative stress markers were observed in the MS group with the sparse exception of a significant decrease of bowel GPx activity when compared to control group. These results should not however overrule the occurrence of oxidative stress in the MS group; instead they may indicate a different oxidative stress mechanism. The reactive oxygen species target not only lipids but also cause protein oxidation and nitration and DNA damage [61], and in this line, recent reports on the effects of maternal separation stress in rats highlight an abnormal elevated nitrosylation profile in the hypothalamus [62], or, in accordance with our results, no differences in antioxidant enzymes activities (SOD, GPx and catalase), but an increased index of DNA breaks in hippocampus [63]. 

The significant correlations between the behavioral indicators and oxidative stress markers, identified predominantly at the brain level, but also a number of weak to moderate correlations between brain and bowel oxidative markers may arguably offer a perspective on the oxidative stress effects: more acutely centrally, where it appears directly correlated to neuropsychiatric affective and cognitive symptoms, as have showed in previous works of our group [64,65], and less pronounced peripherally, in GI tissues, possibly co-occurring along with a low-inflammatory response as supported by other studies [66]. In this way, present results would suggest that GI tissue oxidative damage follows the central nervous system damage, with a similar pattern for oxidant/antioxidant markers, and demonstrate furthermore the connection between brain and gut, described in the literature [67]. 

In summary, the present data show that exposure to combined early and adult life stress including an original sequence of predictable and unpredictable stressors results in significant altered intestinal transit, anxiety and depression-like behaviors and decreased short term cognitive capacity. Its more significant effect compared to either of single stressors was accompanied by increased oxidative stress in colon and predominantly in brain, which suggests the involvement of a neurologic component in the pathogenesis of IBS. Out of the two separate stressors, maternal separation and multifactorial stress, we found only for the latter a similar pattern with the combined stress group in terms of oxidative stress markers dynamics, suggesting that what is driving the effect is mostly the multifactorial stress exposure. This would not rule out the impact of the MS factor on the antioxidant balance, as some oxidative aggravations were observed in its case as well, but may point to different oxidative pathways or even to some degree of inconsistency in the MS protocol as some studies suggested previously [68]. Heterotypical stress exposure could arguably determine a more immediate central response than early-life stress, but, overall, the combination of the two types of stressors reflected more accurately IBS visceral and affective specific symptoms, advocating for a more precise IBS model in mice.

## 5. Conclusions

This study provides additional evidence on the effect of stress exposure on the gastrointestinal and neurological status, in a multifactorial animal model of IBS. The combination of early-life and chronic unpredictable adult-life stress can lead to important depressive and anxiety-like behaviors accompanying alterations of intestinal transit. Oxidative stress may play an important role in IBS development, acting on central and peripheral levels. 

## Figures and Tables

**Figure 1 brainsci-10-00865-f001:**
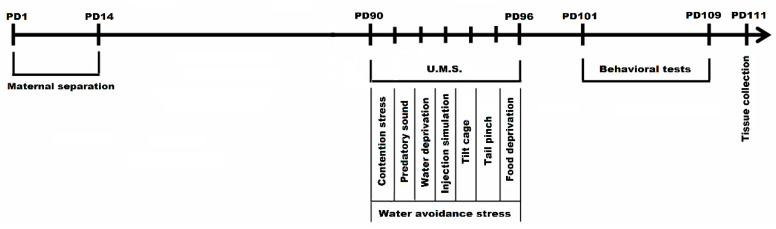
Experimental design of maternal separation and chronic unpredictable mild stress combination in a complex IBS mice model (PD1-PD111 = postnatal days 1-111; U.M.S = unpredictable mild stressors).

**Figure 2 brainsci-10-00865-f002:**
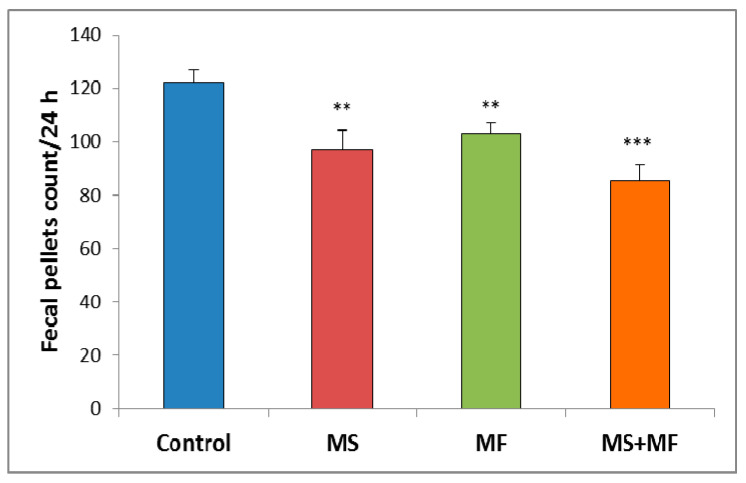
The effect of various combinations of stress factors on gastrointestinal tract habits, as given by fecal pellets count per 24 h. The values are mean ± S.E.M (*n* = 10 per group, ** *p* < 0.01, *** *p* < 0.001, MS = maternal separation, MF = multifactorial stress.

**Figure 3 brainsci-10-00865-f003:**
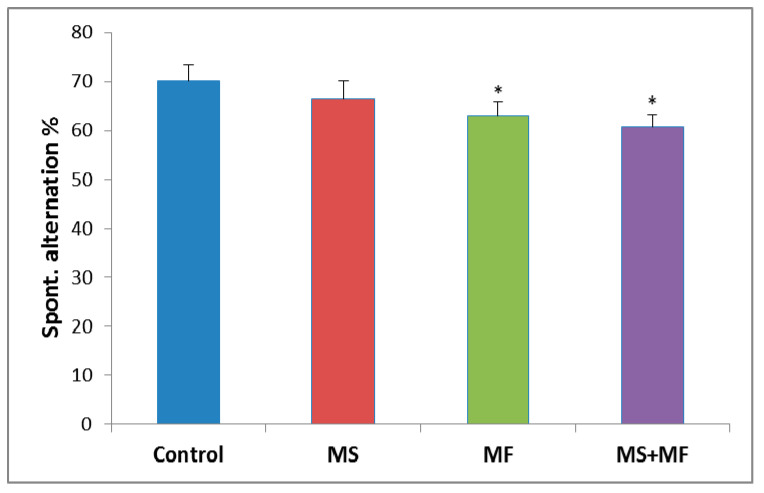
The effects of various combinations of stress factors in the Y-maze test, as showed by the spontaneous alternation parameter. The values are mean ± S.E.M (*n* = 10 per group, * *p* < 0.05 vs. control, MS = maternal separation, MF = multifactorial stress.

**Figure 4 brainsci-10-00865-f004:**
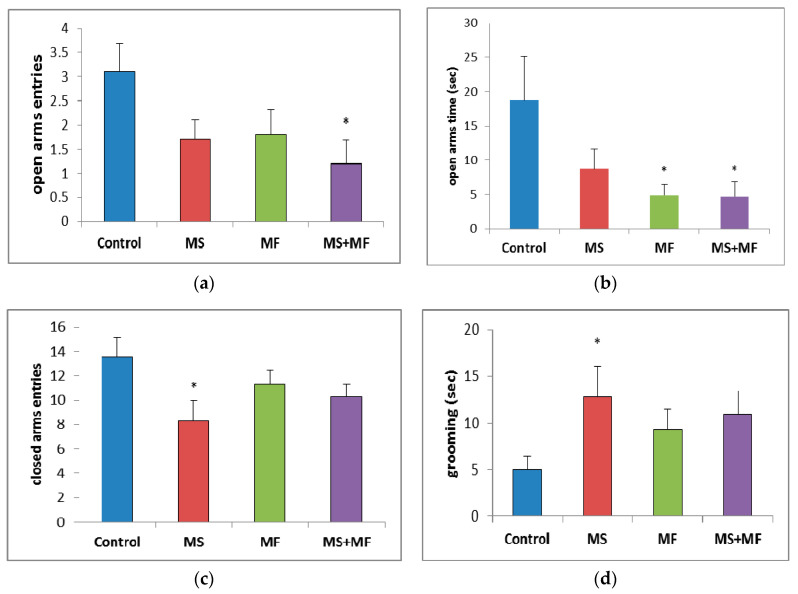
The effects of various combinations of stress factors on the parameters evaluated in elevated plus maze: (**a**) open arms entries, (**b**) open arms time, (**c**) closed arms entries and (**d**) grooming time. The values are mean ± S.E.M (*n* = 10 per group, * *p* < 0.05 vs. control, MS = maternal separation, MF = multifactorial stress.

**Figure 5 brainsci-10-00865-f005:**
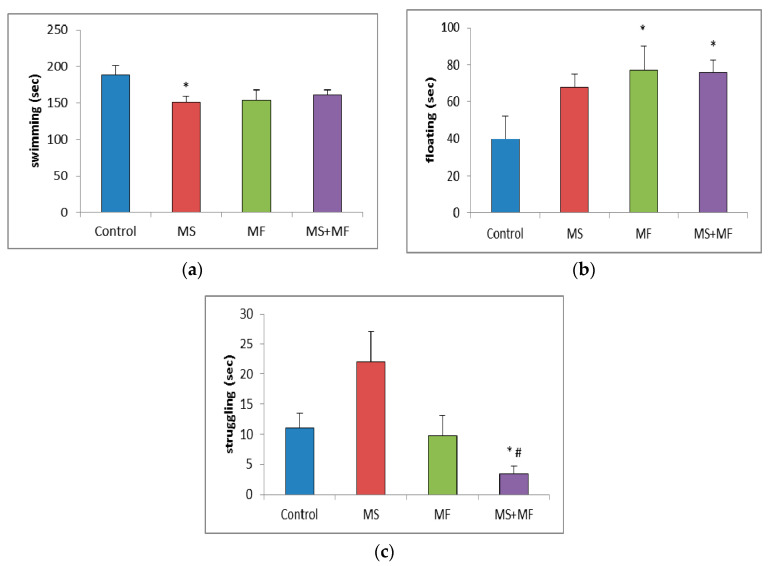
Effect of various combinations of stress factors on the parameters evaluated in the forced swim test: (**a**) swimming, (**b**) floating and (**c**) struggling time (seconds). The values are mean ± S.E.M (*n* = 10 per group, * *p* < 0.05 vs. control and # *p* < 0.05 vs. MS group (MS = maternal separation, MF = multifactorial stress).

**Figure 6 brainsci-10-00865-f006:**
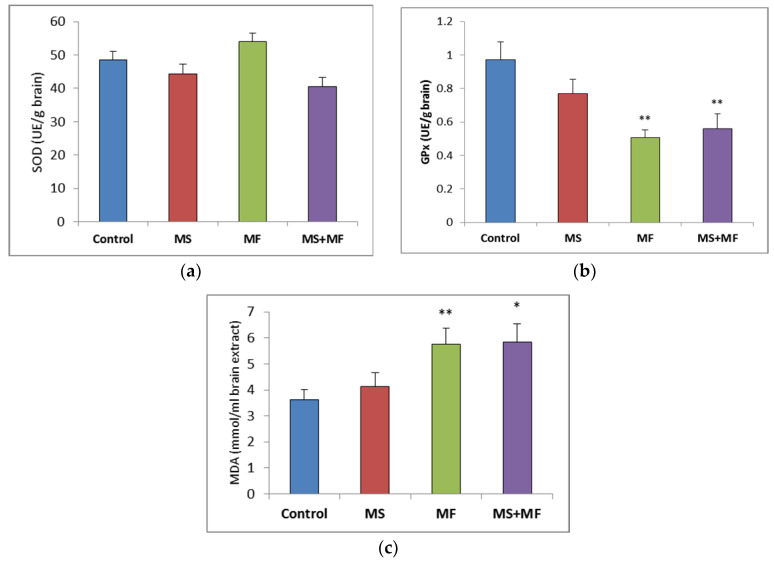
The effect of various combinations of stress factors on oxidative stress markers in the mice brain: (**a**) SOD activity, (**b**) GPx activity and (**c**) MDA levels. The values are mean ± S.E.M. (*n* = 10 per group, * *p* < 0.05, and ** *p* < 0.01, MS = maternal separation, MF = multifactorial stress).

**Figure 7 brainsci-10-00865-f007:**
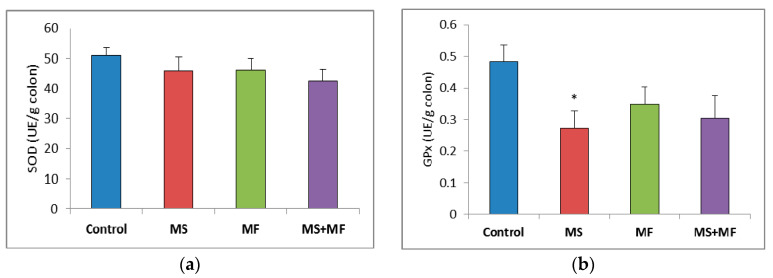
The effect of various combinations of stress factors on oxidative stress markers in the mice bowel tissue: (**a**) SOD activity, (**b**) GPx activity and (**c**) MDA levels. The values are mean ± S.E.M. (*n* = 3 per group), * *p* < 0.05 vs. control, MS = maternal separation, MF = multifactorial stress.

**Figure 8 brainsci-10-00865-f008:**
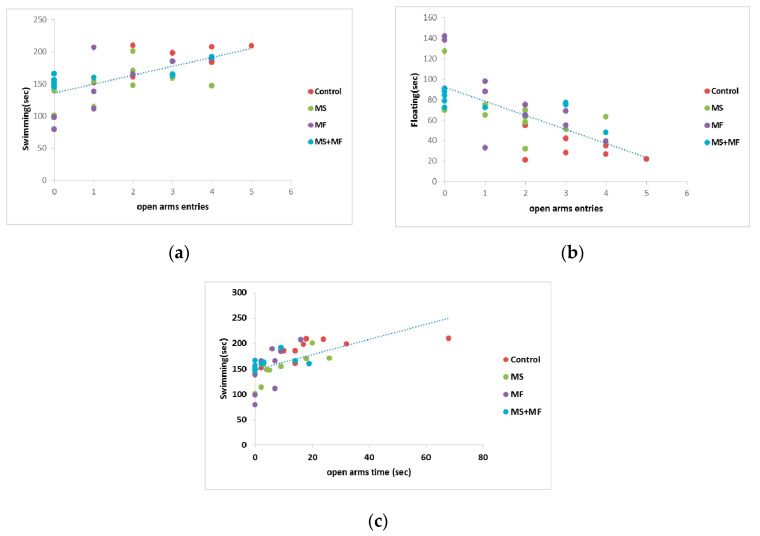
Statistical correlations between behavioral parameters: (**a**) swimming vs. frequency of open arms entries, (**b**) floating duration vs. frequency of open arms entries, (**c**) swimming vs. open arms time (*n* = 40, OAE = open arms entries, OAT = open arms time).

**Figure 9 brainsci-10-00865-f009:**
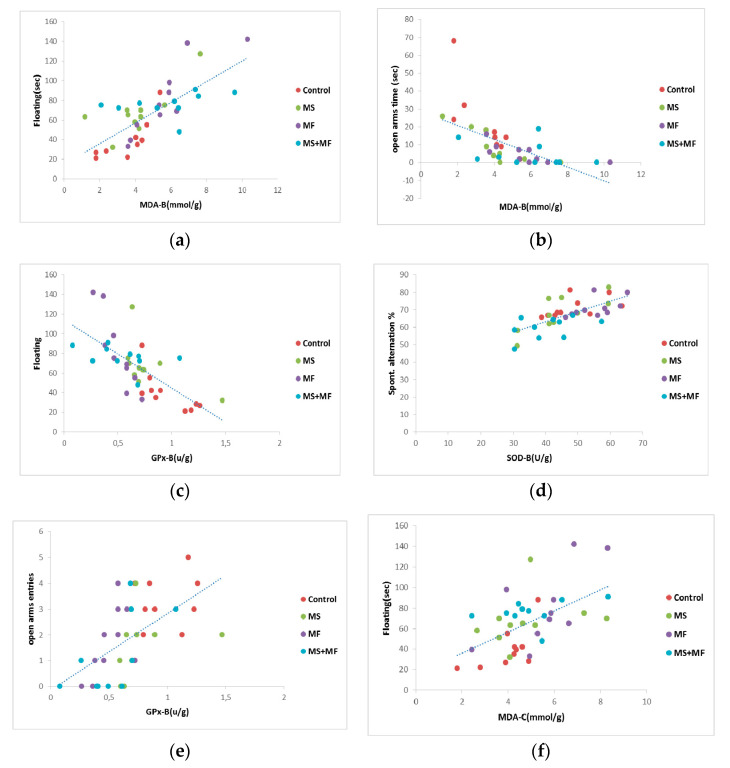
Statistical correlations between behavioral parameters: (**a**) floating duration vs. MDA brain levels, (**b**) open arms time vs. MDA brain levels, (**c**) floating vs. brain GPx activity, (**d**) spontaneous alternation vs. brain SOD activity, (**e**) open arms entries vs. brain GPx activity, (**f**) floating and bowel MDA levels (*n* = 40, OAT = open arms time, OAE = open arms entries, MDA-B = MDA brain level, SOD-B = brain activity of SOD, GPx-B = activity of brain GPx, MDA-C = colon MDA level).

**Figure 10 brainsci-10-00865-f010:**
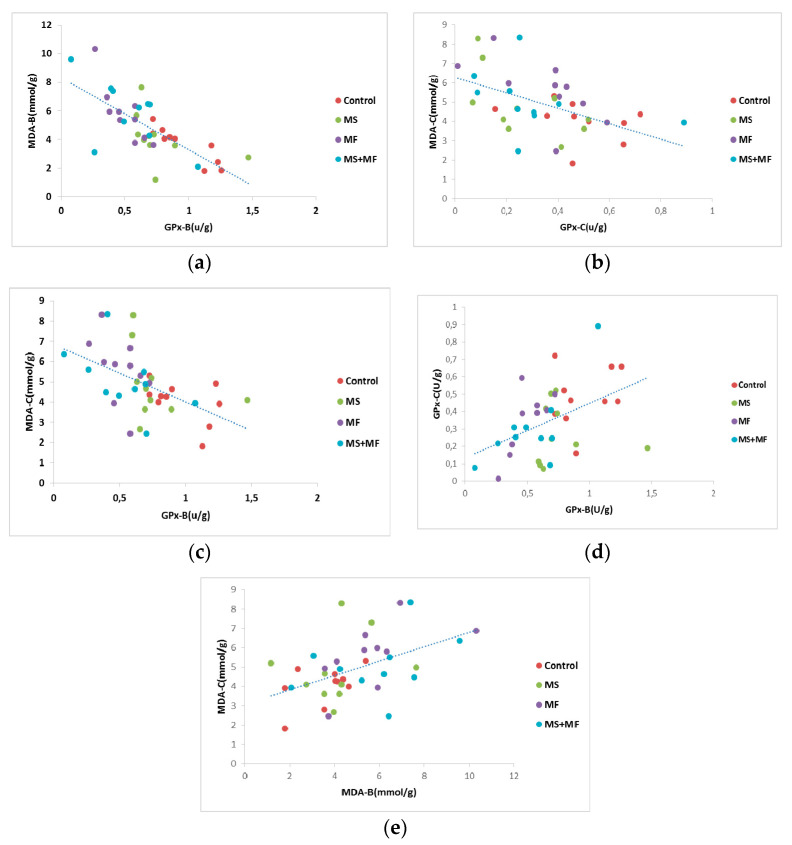
Statistical correlations between oxidative stress parameters in brain and colon: (**a**) brain GPx activity vs. brain MDA levels, (**b**) bowel GPx activity vs. bowel MDA levels, (**c**) brain GPx activity vs. bowel MDA levels, (**d**) brain GPx and bowel GPx activity, (**e**) brain MDA and bowel MDA levels (*n* = 40, MDA-B = MDA brain level, GPx-B = brain GPx activity, GPx-C = colon GPx activity, MDA-C = colon MDA level).

**Table 1 brainsci-10-00865-t001:** The types of stressors applied in each experimental group of mice.

Group 1 (MS)	Group 2 (MF)		Group 3 (MS+MF)		Group Control
maternal separation			maternal separation		
	multifactorial stress		multifactorial stress		
	a. unpredictable	b. repetitive	a. unpredictable	b. repetitive	
	(1) restraint stress	daily water avoidance stress	(1) restraint stress	daily water avoidance stress	
	(2) predator sound		(2) predator sound		
	(3) water deprivation		(3) water deprivation		
	(4) injection simulation		(4) injection simulation		
	(5) tilt cage		(5) tilt cage		
	(6) tail pinch		(6) tail pinch		
	(7) food deprivation		(7) food deprivation		

## Data Availability

All of the data generated and analyzed during this study are included in this published article.

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
