# Peer review of "Behavioral and Oxidative Stress Changes in Mice Subjected to Combinations of Multiple Stressors Relevant to Irritable Bowel Syndrome"

_brainsci, 2020, doi:10.3390/brainsci10110865_

Round 1
Reviewer 1 Report
Cojocariu et al. present a study assessing the effects of stress in producing IBS-like symptoms. The measures used were behavioural indices of depression and anxiety, fecal output, and changes in oxidative stress enzymes in brain and bowel tissue. Four groups of mice were used in a design that combined early-life maternal separation and adult-life multifactorial stress. Overall, they found that relative to the control group, the cumulative effects of early- and late-life stressors (group MS+MF) produced the greatest change in the various measures used, however, there was some variability in this respect depending on the specific measure. The study is overall well-designed and interesting, particularly the finding that MF contributed more than MS to several of the perturbations. My suggestions are focused on improving overall clarity, mostly in the results section. Detailed comments are as follows:
Introduction:
The introduction was overall fine, but I think the last paragraph should be re-phrased. It is written that “[o]ur hypothesis was that cumulative stress comprising of early and adult life stressors would improve the IBS mouse model.” This is not a hypothesis in the way that one typically expects, which is a specific prediction made about the direction of the difference between experimental groups. Instead this sentence refers to the improvement of the IBS mouse model. Can the authors instead provide the experimental hypotheses that are tested in this study? What did they expect would happen and why?
Method:
My suggestions here are mostly minor for improving clarity.
- Please clarify “natural light cycle”.
- The authors used 7 types of unpredictable mild stress. Was the sequence of this counterbalanced across mice or did all mice experience the same order (e.g. did all mice receive contention stress before sound predator)
- For the water avoidance stress, the authors write that the control groups were “placed on the same platform but in a waterless container for 1 hour”. Please clarify that this control group manipulation is not in itself stressful.
- For the forced swim test: can the authors please provide more details as to how the different behaviours were scored, what criteria were used for each behaviour and also how swimming was distinguished from struggling?
Results:
- There is some inconsistency in the reporting of the statistics, sometimes the overall test is reported with F statistic and exact p value indicated, while other times, it is just the approximate p value reported for the follow-up contrast only. Please report all F and T test values, exact p values and confidence intervals or estimates of effect sizes.
- There is also inconsistency in the group names used in-text versus in figures. For example, in 3.1.3., the authors refer to the SC+NMS group, but in the figures there is no such group.
- Please label individual panels of the figures. For example, there are two panels in Figure 4 (3 panels in Figure 6), they should be labelled Figure 4a, Figure 4b etc.
- Section 3.2.: the results for the GPx and MDA levels in the brain are that the MF and MS+MF groups are both significantly different to the control group, but not the MS group. This suggests that what is driving the effect is the MF manipulation. The authors may consider using a follow-up contrast which combines both MF and MS+MF groups compared to the control group.
- Section 3.4.: Please label all x and y axes in the figures. At the moment it is hard for the reader to know which measure is on which axis.
- Please revise for consistency between what is reported in-text and what is shown in the figures. What is reported in-text is the Pearson’s r, while in the figures what is reported is the R2 Please also be consistent with reporting p values.
- Section 3.4.: Consider removing the correlation between swimming and floating as the measures would almost be perfectly orthogonal to each other, since a rat cannot be both floating and swimming at the same time. Hence, the strong negative correlation here simply reflects that instead of anything relating to the experimental manipulations.
- Consider showing correlations for each group rather than collapsed across all groups. At the very least, the authors could indicate in different colours, which data points correspond to which groups in the correlations.
Discussion:
- The authors write that their “initial assumption was confirmed” with regards to GI symptom occurrence, however I don’t think this assumption (or any other) was made explicit in the Introduction. This is related to my comment on the Introduction that the authors should make their experimental hypotheses and predicted results explicit.
- “If we regard struggling as an escape-directed behaviour driven by anxiety state [45], we could presume that the MS stress factor nature could predispose to anxious behavior, whereas an increased stressful load, as in the case of the MS+MF group, would overcome anxiety threshold and shift it to depressive behavior, when struggling significantly decreases and becomes relevant for installation of a depressive state in which individuals no longer try to escape.” The MS group showed significantly less swimming than the controls; please explain how this reflects predisposition to anxious behaviour
- “We have found less significant deficits of short-term memory in the case of maternal separation exposure.” Please clarify what they mean by deficits in short-term memory. Is this about the Y maze?
Author Response
Response to the first reviewer:
We would like to kindly thank the reviewer for the very helpful comments and numerous suggestions, as well as for the very careful reading of the paper.
Cojocariu et al. present a study assessing the effects of stress in producing IBS-like symptoms. The measures used were behavioural indices of depression and anxiety, fecal output, and changes in oxidative stress enzymes in brain and bowel tissue. Four groups of mice were used in a design that combined early-life maternal separation and adult-life multifactorial stress. Overall, they found that relative to the control group, the cumulative effects of early- and late-life stressors (group MS+MF) produced the greatest change in the various measures used, however, there was some variability in this respect depending on the specific measure. The study is overall well-designed and interesting, particularly the finding that MF contributed more than MS to several of the perturbations. My suggestions are focused on improving overall clarity, mostly in the results section. Detailed comments are as follows:
- Introduction:
The introduction was overall fine, but I think the last paragraph should be re-phrased. It is written that “[o]ur hypothesis was that cumulative stress comprising of early and adult life stressors would improve the IBS mouse model.” This is not a hypothesis in the way that one typically expects, which is a specific prediction made about the direction of the difference between experimental groups. Instead this sentence refers to the improvement of the IBS mouse
model. Can the authors instead provide the experimental hypotheses that are tested in this study?
What did they expect would happen and why?
We have rephrased and ellaborated on the hypothesis, as follows:
“In the current study we aimed to investigate the effect of stress exposure on behavior, intestinal transit and oxidative status in mice by using combinations of previously validated IBS stress-based paradigms. Our hypothesis was that the intensity of cumulative stress comprising of early and adult life stressors would alter intestinal motility, affective and cognitive states in mice more than either of the separate stressors. We also assumed that a disturbed antioxidative balance in brain and in colon as a result of psychological stress would accompany and correlate with behavioral changes, which might help to explain the biochemical mechanisms underlying IBS.”
- Method:
My suggestions here are mostly minor for improving clarity. Please clarify “natural light cycle”.
Natural light cycle means that we have opted for natural not artificial illumination; the experimental testing part covered March and April months when days are sensibly equal in length.
- The authors used 7 types of unpredictable mild stress. Was the sequence of this counterbalanced across mice or did all mice experience the same order (e.g. did all mice receive contention stress before sound predator)
This is a difficult question to answer in short; we did not think that counterbalancing stressors should be necessary, or adequate due to the low number of animals, and we opted for the similar sequence exposure; however, due to logistics/techincal reasons, in practice, we had to divide the first two short period stressors between the two MF exposed groups so as to be able to carry out the protocol within the day. Thus, in the first day, the contention/restraint stress was applied to MF group, whilst the MF+MS group was subjected to predator sound, and in the second day the stressors were switched for these two groups; other than this, we were able to apply the stressors in the mentioned order for the two groups.
- For the water avoidance stress, the authors write that the control groups were “placed on the same platform but in a waterless container for 1 hour”. Please clarify that this control group manipulation is not in itself stressful.
We can not state that the manipulation in the case of controls is completely unstressing, however the continuous water exposure is the principal factor of stress in this paradigm, not the positioning on a platform; we have followed the latest well-established method that is cited in numerous other studies (cf. 1,2,3), although, indeed, there are studies in which control animals are kept in cages.
1 - Yamamoto K, Takao T, Nakayama J, Kiuchi H, Okuda H, Fukuhara S, Yoshioka I, Matsuoka Y, Miyagawa Y, Tsujimura A, Nonomura N. Water avoidance stress induces frequency through cyclooxygenase-2 expression: a bladder rat model. Int J Urol. 2012 Feb;19(2):155-62.
2 - Dias, B., Serrão, P., Cruz, F. et al. Effect of Water Avoidance Stress on serum and urinary NGF levels in rats: diagnostic and therapeutic implications for BPS/IC patients. Sci Rep 9, 14113 (2019).
3 - Bradesi S, Schwetz I, Ennes HS, Lamy CM, Ohning G, Fanselow M, Pothoulakis C, McRoberts JA, Mayer EA. Repeated exposure to water avoidance stress in rats: a new model for sustained visceral hyperalgesia. Am J Physiol Gastrointest Liver Physiol. 2005 Jul;289(1):G42-53.
- For the forced swim test: can the authors please provide more details as to how the different behaviours were scored, what criteria were used for each behaviour and also how swimming was distinguished from struggling?
The test was recorded for all mice and the short films were analyzed by the same experienced observer; the three specified indicators are defined in the literature and are different, immobility being when the animal performs no/minimum movement to stay afloat (this should be different than the short pauses after swimming sessions); discriminating between the swimming an struggling is often a challenge. In order to avoid confusions, we have quantified only the evident forms of struggle when the mouse frantically attempts to climb against the walls, which is clearly different than the more fluid horizontal swimming (1) – the difference would be obvious even for an untrained eye.
1- Lam VYY, Raineki C, Takeuchi LE, Ellis L, Woodward TS, Weinberg J. Chronic Stress Alters Behavior in the Forced Swim Test and Underlying Neural Activity in Animals Exposed to Alcohol Prenatally: Sex- and Time-Dependent Effects. Front Behav Neurosci. 2018 Mar 9;12:42.
- Results:
There is some inconsistency in the reporting of the statistics, sometimes the overall test is reported with F statistic and exact p value indicated, while other times, it is just the approximate p value reported for the follow-up contrast only. Please report all F and T test values, exact p values and confidence intervals or estimates of effect sizes.
We report all the data accordingly, although in our opinion in majority of cases p value should be enough.
- There is also inconsistency in the group names used in-text versus in figures. For example, in
3.1.3., the authors refer to the SC+NMS group, but in the figures there is no such group.
Thank you, we have corrected the two inaccuracies, in the text we were refering to the MS group.
- Please label individual panels of the figures. For example, there are two panels in Figure 4 (3 panels in Figure 6), they should be labelled Figure 4a, Figure 4b etc.
This has been addressed throughout the entire Results section.
- Section 3.2.: the results for the GPx and MDA levels in the brain are that the MF and MS+MF groups are both significantly different to the control group, but not the MS group. This suggests that what is driving the effect is the MF manipulation. The authors may consider using a follow-up contrast which combines both MF and MS+MF groups compared to the control group.
|
Test of Homogeneity of Variances |
|
|
|
|
|
|
Levene Statistic |
df1 |
df2 |
Sig. |
|
GPxB |
.866 |
3 |
36 |
.468 |
|
GPxC |
.060 |
3 |
36 |
.980 |
|
MDAB |
.797 |
3 |
36 |
.504 |
|
MDAC |
.603 |
3 |
36 |
.617 |
Levene’s test shows that data have similar variances.
|
Contrast Coefficients |
|
|
|
|
|
Contrast |
Group |
|
|
|
|
Control |
MS |
MF |
MS+MF |
|
|
1 |
-2 |
0 |
1 |
1 |
|
2 |
-2 |
0 |
1 |
1 |
|
Contrast Tests |
|
|
|
|
|
|
|
|
|
|
Contrast |
Value of Contrast |
Std. Error |
t |
df |
Sig. (2-tailed) |
|
GPxB |
Assume equal variances |
1 |
-.8727 |
.17750 |
-4.917 |
36 |
.000 |
|
2 |
-.8727 |
.17750 |
-4.917 |
36 |
.000 |
||
|
Does not assume equal variances |
1 |
-.8727 |
.16717 |
-5.221 |
17.753 |
.000 |
|
|
2 |
-.8727 |
.16717 |
-5.221 |
17.753 |
.000 |
||
|
GPxC |
Assume equal variances |
1 |
-.3165 |
.14474 |
-2.186 |
36 |
.035 |
|
2 |
-.3165 |
.14474 |
-2.186 |
36 |
.035 |
||
|
Does not assume equal variances |
1 |
-.3165 |
.13951 |
-2.268 |
21.042 |
.034 |
|
|
2 |
-.3165 |
.13951 |
-2.268 |
21.042 |
.034 |
||
|
MDAB |
Assume equal variances |
1 |
4.3543 |
1.40877 |
3.091 |
36 |
.004 |
|
2 |
4.3543 |
1.40877 |
3.091 |
36 |
.004 |
||
|
Does not assume equal variances |
1 |
4.3543 |
1.21884 |
3.573 |
26.135 |
.001 |
|
|
2 |
4.3543 |
1.21884 |
3.573 |
26.135 |
.001 |
||
|
MDAC |
Assume equal variances |
1 |
2.6075 |
1.17058 |
2.227 |
36 |
.032 |
|
2 |
2.6075 |
1.17058 |
2.227 |
36 |
.032 |
||
|
Does not assume equal variances |
1 |
2.6075 |
.96385 |
2.705 |
25.385 |
.012 |
|
|
2 |
2.6075 |
.96385 |
2.705 |
25.385 |
.012 |
Contrast test raveled that having MS or MF+MS decreased the value of GPxB compared to the control group, t(36)= -4.917, p=,000.
Contrast test raveled that having MS or MF+MS decreased the value of GPxC compared to the control group, t(36)= -2.186, p=.035.
Contrast test raveled that having MS or MF+MS increased the value of MDAB compared to the control group, t(36)= 3.091, p=.004.
Contrast test raveled that having MS or MF+MS increased the value of MDAC compared to the control group, t(36)= 2.227, p=.032.
- Section 3.4.: Please label all x and y axes in the figures. At the moment it is hard for the reader to know which measure is on which axis.
We have labeled the axes.
- Please revise for consistency between what is reported in-text and what is shown in the figures. What is reported in-text is the Pearson’s r, while in the figures what is reported is the R2
Indeed, there is a difference between them, for consistency, we have reported now only in text R.
- Please also be consistent with reporting p values.
Thank you for the indication
- Section 3.4.: Consider removing the correlation between swimming and floating as the measures would almost be perfectly orthogonal to each other, since a rat cannot be both floating and swimming at the same time. Hence, the strong negative correlation here simply reflects that instead of anything relating to the experimental manipulations.
We have removed the correlation.
- Consider showing correlations for each group rather than collapsed across all groups. At the very least, the authors could indicate in different colours, which data points correspond to which
groups in the correlations.
Correlations for each group would add too many graphical representations and, although being more detailed, would not substantially support/change our general argumentation. We have changed the colours between the groups, as this was the only option our software currently allowed us; we would require a few more days to make adequate changes.
- Discussion: The authors write that their “initial assumption was confirmed” with regards to GI symptom occurrence, however I don’t think this assumption (or any other) was made explicit in the Introduction. This is related to my comment on the Introduction that the authors should make their experimental hypotheses and predicted results explicit.
We have developed some more the final part of the discussions as follows: “In summary, the present data show that exposure to combined early and adult life stress including an original sequence of predictable and unpredictable stressors results in significant altered intestinal tranzit, anxiety and depression-like behaviors and decreased short term cognitive capacity. Its more significant effect compared to either of single stressors was accompanied by increased oxidative stress in colon and predominantly in brain, which suggests the involvement of a neurologic component in the pathogenesis of IBS. Out of the two separate stressors, maternal separation and multifactorial stress, we found only for the latter a similar pattern with the combined stress group in terms of oxidative stress markers dynamics, suggesting that what is driving the effect is mostly the multifactorial stress exposure. This would not rule out the impact of the MS factor on the antioxidant balance, as some oxidative aggravations were observed in its case as well, but may point to different oxidative pathways or even to some degree of inconsistency in the MS protocol as some studies suggested previously [70]. Heterotypical stress exposure could arguably determine a more immediate central response than early-life stress, but, overall, the combination of the two types of stressors reflected more accurately IBS visceral and affective specific symptoms, advocating for a more precise IBS model in mice.”
- “If we regard struggling as an escape-directed behaviour driven by anxiety state [45], we could presume that the MS stress factor nature could predispose to anxious behavior, whereas an increased stressful load, as in the case of the MS+MF group, would overcome anxiety threshold and shift it to depressive behavior, when struggling significantly decreases and becomes relevant for installation of a depressive state in which individuals no longer try to escape.”
The MS group showed significantly less swimming than the controls; please explain how this reflects predisposition to anxious behaviour.
The forced swimming test is accepted as a depression assessement tool, less swimming and especially more floating are generally viewed as reflecting installement of depressive states. The MS group displayed signifcant less swimming, but not as significantly more floating, when compared to controls, this would in our opinion reflect some milder depressive state (if any), unlike in the other two stressed groups; yet, there is more struggling which coupled with more floating led to significantly decreased swimming time, which could indicate a different reactivity to the external stimuli than normal. Struggling significance is not clearly acknowledged, sometimes regarded as an extension of swimming, sometimes as a coping strategy with the escapeless environment, but also, it may was described by [43-45], as cited in our paper, as the manifestation of heightened anxiety state; we assumed that this group would generally react in a more anxious way, due to this preponderance of struggling/climbing manifestations.
- “We have found less significant deficits of short-term memory in the case of maternal separation exposure.” Please clarify what they mean by deficits in short-term memory. Is this about the Y maze?
Indeed, these deficits in memory were assessed by the Y maze test. We have addressed this in the text, as follows: “We have found less significant deficits in short-term memory during the Y maze task in the case of maternal separation exposure”
Reviewer 2 Report
This paper studied the behavioral as well as the biochemical effects of multiple combinations of stressors on mice. The Paper is well written, and the results are well presented and discussed.
Minor comments:
- Please stick to a unified way of reference citation when you cite multiple references [1] [2] => [1, 2].
- In the methods section, it is a bit difficult to understand the whole idea. I suggest creating a table that explains the mice groups and what kind of stressors they were exposed to.
- One way ANOVA is not the best statistical test for this kind of data. I suggest trying the two-way ANOVA. I believe it can add more detailed information.
- Judging from some figures and the significance marks, some figures probably used standard deviation and not the standard error of the mean. Please take a second look and confirm that.
- Some figures are not well aligned.
- Please stress the novel and significant findings your paper has provided in both the introduction and conclusion.
Author Response
Response to the second reviewer
We would like to kindly thank the reviewer for the rigorously and precise comments, clearly delineating the weaknesses and offering the essential indications to remediate these.
- This paper studied the behavioral as well as the biochemical effects of multiple combinations of stressors on mice. The Paper is well written, and the results are well presented and discussed.
Minor comments: Please stick to a unified way of reference citation when you cite multiple references [1] [2] => [1, 2].
Thank you for pointing this slip, we have changed all citations to the “[1,2]” system.
- In the methods section, it is a bit difficult to understand the whole idea. I suggest creating a table that explains the mice groups and what kind of stressors they were exposed to.
Thank you for the suggestion; we have added this table in the Methods:
|
Group 1 (MS)
|
Group 2 (MF) |
Group 3 (MS+MF) |
Group control |
||
|
maternal separation |
|
maternal separation |
|
||
|
|
multifactorial stress |
multifactorial stress |
|
||
|
|
a.unpredictable |
b.repetitive |
a.unpredictable |
b.repetitive |
|
|
(1) restraint stress |
daily water avoidance stress |
(1) restraint stress |
daily water avoidance stress |
||
|
(2) predator sound |
(2) predator sound |
||||
|
(3) water deprivation |
(3) water deprivation |
||||
|
(4) injection simulation |
(4) injection simulation |
||||
|
(5) tilt cage |
(5) tilt cage |
||||
|
(6) tail pinch |
(6) tail pinch |
||||
|
(7) food deprivation |
(7) food deprivation |
||||
- One way ANOVA is not the best statistical test for this kind of data. I suggest trying the two-way ANOVA. I believe it can add more detailed information.
We have considered one way ANOVA because we have only one single independent factor in the data which is the group membership, thus we are comparing groups to establish whether there is relationship between them, within each group there are observations.
|
|
|
|
|
|
ctr vs MS |
|
|
|
|
|
|
|
|
|
|
|
|
|
|
GROUP |
SA% |
OAE |
OAT |
CAE |
Grooming |
Swimming |
Floating |
Struggling |
|
1 |
72,22222 |
4 |
24 |
10 |
8 |
208 |
27 |
5 |
|
2 |
68,25 |
2 |
68 |
7 |
11 |
210 |
21 |
9 |
|
3 |
67,64706 |
3 |
14 |
16 |
5 |
185 |
42 |
13 |
|
4 |
66,62309 |
4 |
9 |
18 |
4 |
184 |
39 |
17 |
|
5 |
73,81818 |
3 |
17 |
9 |
8 |
198 |
42 |
0 |
|
6 |
65,75758 |
2 |
14 |
15 |
0 |
161 |
55 |
24 |
|
7 |
66,66667 |
1 |
2 |
16 |
12 |
152 |
88 |
0 |
|
8 |
80 |
5 |
18 |
8 |
0 |
209 |
22 |
9 |
|
9 |
81,25 |
3 |
32 |
15 |
2 |
199 |
28 |
13 |
|
10 |
68,34848 |
4 |
10 |
22 |
0 |
185 |
35 |
20 |
|
1 |
76,92308 |
2 |
20 |
3 |
19 |
201 |
32 |
7 |
|
2 |
82,75333 |
2 |
18 |
4 |
23 |
170 |
70 |
0 |
|
3 |
73,33333 |
2 |
26 |
2 |
33 |
171 |
63 |
6 |
|
4 |
76,28565 |
3 |
3 |
7 |
18 |
159 |
51 |
30 |
|
5 |
68,10747 |
1 |
9 |
10 |
0 |
154 |
65 |
21 |
|
6 |
66,66667 |
2 |
4 |
6 |
3 |
148 |
58 |
34 |
|
7 |
62,78035 |
4 |
5 |
10 |
8 |
147 |
63 |
30 |
|
8 |
61,76471 |
0 |
0 |
12 |
12 |
140 |
70 |
30 |
|
9 |
58,06452 |
1 |
2 |
20 |
2 |
114 |
75 |
51 |
|
10 |
49,33333 |
0 |
0 |
9 |
10 |
101 |
127 |
12 |
|
|
|
|
|
|
|
|
|
|
|
|
|
|
|
|
|
|
|
|
|
|
Analyse de variance: deux facteurs avec répétition d'expérience |
|
||||||
|
|
|
|
|
|
|
|
|
|
|
|
RAPPORT DÉTAILLÉ |
OAE |
OAT |
CAE |
Grooming |
Swimming |
Floating |
|
|
|
72,2222 |
|
|
|
|
|
|
|
|
|
Nombre d'échantillons |
10 |
10 |
10 |
10 |
10 |
10 |
|
|
|
Somme |
31 |
208 |
136 |
50 |
1891 |
399 |
|
|
|
Moyenne |
3,1 |
20,8 |
13,6 |
5 |
189,1 |
39,9 |
|
|
|
Variance |
1,433333 |
343,0667 |
23,82222 |
20,88889 |
399,2111 |
395,6556 |
|
|
|
|
|
|
|
|
|
|
|
|
|
76,9231 |
|
|
|
|
|
|
|
|
|
Nombre d'échantillons |
10 |
10 |
10 |
10 |
10 |
10 |
|
|
|
Somme |
17 |
87 |
83 |
128 |
1505 |
674 |
|
|
|
Moyenne |
1,7 |
8,7 |
8,3 |
12,8 |
150,5 |
67,4 |
|
|
|
Variance |
1,566667 |
86,45556 |
27,78889 |
109,5111 |
816,2778 |
586,4889 |
|
|
|
|
|
|
|
|
|
|
|
|
|
Total |
|
|
|
|
|
|
|
|
|
Nombre d'échantillons |
20 |
20 |
20 |
20 |
20 |
20 |
|
|
|
Somme |
48 |
295 |
219 |
178 |
3396 |
1073 |
|
|
|
Moyenne |
2,4 |
14,75 |
10,95 |
8,9 |
169,8 |
53,65 |
|
|
|
Variance |
1,936842 |
241,9868 |
31,83947 |
77,77895 |
967,8526 |
664,2395 |
|
|
|
|
|
|
|
|
|
|
|
|
|
|
|
|
|
|
|
|
|
|
|
ANALYSE DE VARIANCE |
|
|
|
|
|
||
|
|
Source des variations |
Somme des carrés |
Degré de liberté |
Moyenne des carrés |
F |
Probabilité |
Valeur critique pour F |
|
|
|
Échantillon |
86,42857 |
1 |
86,42857 |
0,387098 |
0,534953 |
3,916325 |
|
|
|
Colonnes |
428908,3 |
6 |
71484,71 |
320,1673 |
9,7E-74 |
2,171309 |
|
|
|
Interaction |
12947,17 |
6 |
2157,862 |
9,664678 |
9,69E-09 |
2,171309 |
|
|
|
A l'intérieur du groupe |
28132,4 |
126 |
223,273 |
|
|
|
|
|
|
|
|
|
|
|
|
|
|
|
|
Total |
470074,3 |
139 |
|
|
|
|
|
|
|
|
|
|
ctr vs MF |
|
|
|
|
|
|
|
|
|
|
|
|
|
|
|
GROUP |
SA% |
OAE |
OAT |
CAE |
Grooming |
Swimming |
Floating |
Struggling |
|
1 |
72,22222 |
4 |
24 |
10 |
8 |
208 |
27 |
5 |
|
2 |
68,25 |
2 |
68 |
7 |
11 |
210 |
21 |
9 |
|
3 |
67,64706 |
3 |
14 |
16 |
5 |
185 |
42 |
13 |
|
4 |
66,62309 |
4 |
9 |
18 |
4 |
184 |
39 |
17 |
|
5 |
73,81818 |
3 |
17 |
9 |
8 |
198 |
42 |
0 |
|
6 |
65,75758 |
2 |
14 |
15 |
0 |
161 |
55 |
24 |
|
7 |
66,66667 |
1 |
2 |
16 |
12 |
152 |
88 |
0 |
|
8 |
80 |
5 |
18 |
8 |
0 |
209 |
22 |
9 |
|
9 |
81,25 |
3 |
32 |
15 |
2 |
199 |
28 |
13 |
|
10 |
68,34848 |
4 |
10 |
22 |
0 |
185 |
35 |
20 |
|
1 |
81,25 |
3 |
9 |
14 |
8 |
185 |
55 |
0 |
|
2 |
72,22222 |
1 |
16 |
10 |
7 |
207 |
33 |
0 |
|
3 |
80 |
4 |
6 |
9 |
8 |
189 |
39 |
12 |
|
4 |
68,34848 |
2 |
2 |
18 |
6 |
165 |
65 |
10 |
|
5 |
69,62309 |
3 |
2 |
5 |
0 |
164 |
69 |
7 |
|
6 |
70,81818 |
2 |
7 |
9 |
8 |
165 |
75 |
0 |
|
7 |
67,64706 |
1 |
7 |
10 |
10 |
111 |
98 |
31 |
|
8 |
68,25 |
1 |
0 |
15 |
4 |
138 |
88 |
14 |
|
9 |
66,66667 |
0 |
0 |
12 |
19 |
79 |
138 |
23 |
|
10 |
65,75758 |
0 |
0 |
11 |
23 |
98 |
142 |
0 |
|
|
|
|
|
|
|
|
|
|
|
|
|
|
|
|
|
|
|
|
|
|
|
|
|
|
|
|
|
|
|
|
|
|
|
|
|
|
|
|
|
|
Analyse de variance: deux facteurs avec répétition d'expérience |
|
||||||
|
|
|
|
|
|
|
|
|
|
|
|
RAPPORT DÉTAILLÉ |
OAE |
OAT |
CAE |
Grooming |
Swimming |
Floating |
|
|
|
72,2222 |
|
|
|
|
|
|
|
|
|
Nombre d'échantillons |
10 |
10 |
10 |
10 |
10 |
10 |
|
|
|
Somme |
31 |
208 |
136 |
50 |
1891 |
399 |
|
|
|
Moyenne |
3,1 |
20,8 |
13,6 |
5 |
189,1 |
39,9 |
|
|
|
Variance |
1,433333 |
343,0667 |
23,82222 |
20,88889 |
399,2111 |
395,6556 |
|
|
|
|
|
|
|
|
|
|
|
|
|
81,25 |
|
|
|
|
|
|
|
|
|
Nombre d'échantillons |
10 |
10 |
10 |
10 |
10 |
10 |
|
|
|
Somme |
17 |
49 |
113 |
93 |
1501 |
802 |
|
|
|
Moyenne |
1,7 |
4,9 |
11,3 |
9,3 |
150,1 |
80,2 |
|
|
|
Variance |
1,788889 |
26,54444 |
13,34444 |
46,45556 |
1783,433 |
1386,844 |
|
|
|
|
|
|
|
|
|
|
|
|
|
Total |
|
|
|
|
|
|
|
|
|
Nombre d'échantillons |
20 |
20 |
20 |
20 |
20 |
20 |
|
|
|
Somme |
48 |
257 |
249 |
143 |
3392 |
1201 |
|
|
|
Moyenne |
2,4 |
12,85 |
12,45 |
7,15 |
169,6 |
60,05 |
|
|
|
Variance |
2,042105 |
241,6079 |
18,99737 |
36,76579 |
1434,147 |
1271,734 |
|
|
|
|
|
|
|
|
|
|
|
|
|
|
|
|
|
|
|
|
|
|
|
ANALYSE DE VARIANCE |
|
|
|
|
|
||
|
|
Source des variations |
Somme des carrés |
Degré de liberté |
Moyenne des carrés |
F |
Probabilité |
Valeur critique pour F |
|
|
|
Échantillon |
167,2071 |
1 |
167,2071 |
0,506439 |
0,478 |
3,916325 |
|
|
|
Colonnes |
441250,1 |
6 |
73541,68 |
222,7438 |
1,46E-64 |
2,171309 |
|
|
|
Interaction |
16959,44 |
6 |
2826,574 |
8,561154 |
8,3E-08 |
2,171309 |
|
|
|
A l'intérieur du groupe |
41600,5 |
126 |
330,1627 |
|
|
|
|
|
|
|
|
|
|
|
|
|
|
|
|
Total |
499977,2 |
139 |
|
|
|
|
|
|
|
|
|
|
ctr vs MS+MF |
|
|
|
|
|
|
|
|
|
|
|
|
|
|
|
GROUP |
SA% |
OAE |
OAT |
CAE |
Grooming |
Swimming |
Floating |
Struggling |
|
1 |
72,22222 |
4 |
24 |
10 |
8 |
208 |
27 |
5 |
|
2 |
68,25 |
2 |
68 |
7 |
11 |
210 |
21 |
9 |
|
3 |
67,64706 |
3 |
14 |
16 |
5 |
185 |
42 |
13 |
|
4 |
66,62309 |
4 |
9 |
18 |
4 |
184 |
39 |
17 |
|
5 |
73,81818 |
3 |
17 |
9 |
8 |
198 |
42 |
0 |
|
6 |
65,75758 |
2 |
14 |
15 |
0 |
161 |
55 |
24 |
|
7 |
66,66667 |
1 |
2 |
16 |
12 |
152 |
88 |
0 |
|
8 |
80 |
5 |
18 |
8 |
0 |
209 |
22 |
9 |
|
9 |
81,25 |
3 |
32 |
15 |
2 |
199 |
28 |
13 |
|
10 |
68,34848 |
4 |
10 |
22 |
0 |
185 |
35 |
20 |
|
1 |
63,33333 |
3 |
14 |
14 |
9 |
165 |
75 |
0 |
|
2 |
66,66667 |
1 |
19 |
7 |
4 |
160 |
72 |
8 |
|
3 |
47,22222 |
4 |
9 |
8 |
12 |
192 |
48 |
0 |
|
4 |
64,34024 |
3 |
3 |
13 |
9 |
163 |
77 |
0 |
|
5 |
62,9747 |
0 |
0 |
10 |
28 |
152 |
79 |
9 |
|
6 |
60 |
0 |
0 |
11 |
8 |
166 |
72 |
2 |
|
7 |
53,84615 |
0 |
0 |
7 |
0 |
149 |
91 |
0 |
|
8 |
54,03443 |
0 |
0 |
16 |
13 |
156 |
84 |
0 |
|
9 |
65,36569 |
0 |
0 |
8 |
20 |
145 |
88 |
7 |
|
10 |
58,27027 |
1 |
2 |
9 |
6 |
160 |
72 |
8 |
|
Analyse de variance: deux facteurs avec répétition d'expérience |
|
|
||||||
|
|
|
|
|
|
|
|
|
|
|
RAPPORT DÉTAILLÉ |
OAE |
OAT |
CAE |
Grooming |
Swimming |
Floating |
Struggling |
Total |
|
72,2222 |
|
|
|
|
|
|
|
|
|
Nombre d'échantillons |
10 |
10 |
10 |
10 |
10 |
10 |
10 |
70 |
|
Somme |
31 |
208 |
136 |
50 |
1891 |
399 |
110 |
2825 |
|
Moyenne |
3,1 |
20,8 |
13,6 |
5 |
189,1 |
39,9 |
11 |
40,35714 |
|
Variance |
1,433333 |
343,0667 |
23,82222 |
20,88889 |
399,2111 |
395,6556 |
64,44444 |
4035,769 |
|
|
|
|
|
|
|
|
|
|
|
63,3333 |
|
|
|
|
|
|
|
|
|
Nombre d'échantillons |
10 |
10 |
10 |
10 |
10 |
10 |
10 |
70 |
|
Somme |
12 |
47 |
103 |
109 |
1608 |
758 |
34 |
2671 |
|
Moyenne |
1,2 |
4,7 |
10,3 |
10,9 |
160,8 |
75,8 |
3,4 |
38,15714 |
|
Variance |
2,4 |
47,78889 |
9,788889 |
65,21111 |
168,1778 |
141,7333 |
16,26667 |
3199,526 |
|
|
|
|
|
|
|
|
|
|
|
Total |
|
|
|
|
|
|
|
|
|
Nombre d'échantillons |
20 |
20 |
20 |
20 |
20 |
20 |
20 |
|
|
Somme |
43 |
255 |
239 |
159 |
3499 |
1157 |
144 |
|
|
Moyenne |
2,15 |
12,75 |
11,95 |
7,95 |
174,95 |
57,85 |
7,2 |
|
|
Variance |
2,765789 |
253,3553 |
18,78684 |
49,94474 |
479,5237 |
593,7132 |
53,43158 |
|
|
|
|
|
|
|
|
|
|
|
|
|
|
|
|
|
|
|
|
|
|
ANALYSE DE VARIANCE |
|
|
|
|
|
|
||
|
Source des variations |
Somme des carrés |
Degré de liberté |
Moyenne des carrés |
F |
Probabilité |
Valeur critique pour F |
|
|
|
Échantillon |
169,4 |
1 |
169,4 |
1,39515 |
0,239762 |
3,916325 |
|
|
|
Colonnes |
471825,8 |
6 |
78637,64 |
647,6464 |
4,01E-92 |
2,171309 |
|
|
|
Interaction |
12110,5 |
6 |
2018,417 |
16,62334 |
4,7E-14 |
2,171309 |
|
|
|
A l'intérieur du groupe |
15299 |
126 |
121,4206 |
|
|
|
|
|
|
|
|
|
|
|
|
|
|
|
|
Total |
499404,7 |
139 |
|
|
|
|
|
|
|
|
|
ctr vs MS |
|
|
|
|
|
|
|
|
|
|
|
|
|
GROUP |
SOD-B |
SOD-C |
GPx-B |
GPx-C |
MDA-B |
MDA-C |
|
1 |
63,74616 |
51,85262 |
1,258829 |
0,657872 |
1,802215 |
3,890363 |
|
2 |
44,76248 |
57,59533 |
1,125189 |
0,457872 |
1,798677 |
1,802545 |
|
3 |
53,76248 |
45,81132 |
0,813979 |
0,357872 |
4,034231 |
4,269943 |
|
4 |
42,8869 |
51,79693 |
0,725189 |
0,721257 |
4,39056 |
4,357405 |
|
5 |
49,97569 |
70,64803 |
0,894174 |
0,157257 |
4,025063 |
4,627405 |
|
6 |
38,81047 |
48,57686 |
0,796642 |
0,520876 |
4,650167 |
3,981153 |
|
7 |
40,74616 |
48,39999 |
0,725189 |
0,385143 |
5,398247 |
5,290618 |
|
8 |
59,69098 |
41,43127 |
1,179664 |
0,656643 |
3,568593 |
2,794091 |
|
9 |
47,5936 |
48,35492 |
1,228214 |
0,457872 |
2,385463 |
4,908817 |
|
10 |
43,5936 |
45,59613 |
0,852788 |
0,463191 |
4,131877 |
4,257405 |
|
1 |
45,00292 |
48,7484 |
1,468829 |
0,188403 |
2,745316 |
4,084283 |
|
2 |
59,62048 |
72,63902 |
0,892141 |
0,209469 |
3,560715 |
3,617263 |
|
3 |
59,46695 |
58,85149 |
0,741763 |
0,388403 |
1,17457 |
5,18433 |
|
4 |
41,03651 |
57,26103 |
0,691763 |
0,501435 |
4,221828 |
3,613932 |
|
5 |
49,8827 |
49,54426 |
0,698755 |
0,243041 |
3,588182 |
4,647501 |
|
6 |
41,14155 |
37,79614 |
0,652788 |
0,414353 |
3,97582 |
2,651837 |
|
7 |
42,50869 |
40,41264 |
0,732738 |
0,51943 |
4,312656 |
4,093445 |
|
8 |
41,15615 |
30,50961 |
0,604141 |
0,088853 |
4,321651 |
8,277756 |
|
9 |
31,41876 |
39,60889 |
0,594175 |
0,109469 |
5,657252 |
7,295002 |
|
10 |
31,25903 |
23,06196 |
0,631763 |
0,067789 |
7,645848 |
4,987947 |
|
|
|
|
|
|
|
|
|
|
|
|
|
|
|
|
|
|
|
|
|
|
|
|
|
|
|
|
|
|
|
|
|
Analyse de variance: deux facteurs avec répétition d'expérience |
||||||
|
|
|
|
|
|
|
|
|
RAPPORT DÉTAILLÉ |
SOD-C |
GPx-B |
GPx-C |
MDA-B |
MDA-C |
Total |
|
63,7462 |
|
|
|
|
|
|
|
Nombre d'échantillons |
10 |
10 |
10 |
10 |
10 |
50 |
|
Somme |
510,0634 |
9,599856 |
4,835856 |
36,18509 |
40,17974 |
600,8639 |
|
Moyenne |
51,00634 |
0,959986 |
0,483586 |
3,618509 |
4,017974 |
12,01728 |
|
Variance |
66,46867 |
0,045654 |
0,027992 |
1,507012 |
1,049368 |
402,4804 |
|
|
|
|
|
|
|
|
|
45,0029 |
|
|
|
|
|
|
|
Nombre d'échantillons |
10 |
10 |
10 |
10 |
10 |
50 |
|
Somme |
458,4334 |
7,708854 |
2,730645 |
41,20384 |
48,4533 |
558,5301 |
|
Moyenne |
45,84334 |
0,770885 |
0,273064 |
4,120384 |
4,84533 |
11,1706 |
|
Variance |
214,1059 |
0,067547 |
0,029007 |
2,896069 |
2,993029 |
350,3891 |
|
|
|
|
|
|
|
|
|
Total |
|
|
|
|
|
|
|
Nombre d'échantillons |
20 |
20 |
20 |
20 |
20 |
|
|
Somme |
968,4968 |
17,30871 |
7,5665 |
77,38893 |
88,63304 |
|
|
Moyenne |
48,42484 |
0,865435 |
0,378325 |
3,869446 |
4,431652 |
|
|
Variance |
139,9186 |
0,063032 |
0,038662 |
2,151954 |
2,094956 |
|
|
|
|
|
|
|
|
|
|
|
|
|
|
|
|
|
|
ANALYSE DE VARIANCE |
|
|
|
|
||
|
Source des variations |
Somme des carrés |
Degré de liberté |
Moyenne des carrés |
F |
Probabilité |
Valeur critique pour F |
|
Échantillon |
17,92158 |
1 |
17,92158 |
0,619716 |
0,433221 |
3,946876 |
|
Colonnes |
34167,45 |
4 |
8541,862 |
295,3717 |
7,56E-51 |
2,472927 |
|
Interaction |
120,4435 |
4 |
30,11087 |
1,041213 |
0,390511 |
2,472927 |
|
A l'intérieur du groupe |
2602,712 |
90 |
28,91902 |
|
|
|
|
|
|
|
|
|
|
|
|
Total |
36908,52 |
99 |
|
|
|
|
|
|
|
ctr vs MF |
|
|
|
|
|
|
|
|
|
|
|
|
|
GROUP |
SOD-B |
SOD-C |
GPx-B |
GPx-C |
MDA-B |
MDA-C |
|
1 |
63,74616 |
51,85262 |
1,258829 |
0,657872 |
1,802215 |
3,890363 |
|
2 |
44,76248 |
57,59533 |
1,125189 |
0,457872 |
1,798677 |
1,802545 |
|
3 |
53,76248 |
45,81132 |
0,813979 |
0,357872 |
4,034231 |
4,269943 |
|
4 |
42,8869 |
51,79693 |
0,725189 |
0,721257 |
4,39056 |
4,357405 |
|
5 |
49,97569 |
70,64803 |
0,894174 |
0,157257 |
4,025063 |
4,627405 |
|
6 |
38,81047 |
48,57686 |
0,796642 |
0,520876 |
4,650167 |
3,981153 |
|
7 |
40,74616 |
48,39999 |
0,725189 |
0,385143 |
5,398247 |
5,290618 |
|
8 |
59,69098 |
41,43127 |
1,179664 |
0,656643 |
3,568593 |
2,794091 |
|
9 |
47,5936 |
48,35492 |
1,228214 |
0,457872 |
2,385463 |
4,908817 |
|
10 |
43,5936 |
45,59613 |
0,852788 |
0,463191 |
4,131877 |
4,257405 |
|
1 |
55,07245 |
41,03772 |
0,657872 |
0,407347 |
4,10277 |
5,284317 |
|
2 |
63,06196 |
51,30587 |
0,726023 |
0,498766 |
3,593145 |
4,934194 |
|
3 |
65,38467 |
26,00163 |
0,581285 |
0,392298 |
3,753145 |
2,436559 |
|
4 |
49,54316 |
41,03772 |
0,581332 |
0,390919 |
5,387535 |
6,638686 |
|
5 |
52,16116 |
40,69763 |
0,580521 |
0,434719 |
6,324133 |
5,789718 |
|
6 |
58,38045 |
51,23371 |
0,463191 |
0,38923 |
5,342263 |
5,864535 |
|
7 |
48,16599 |
40,75431 |
0,457872 |
0,592298 |
5,932739 |
3,936318 |
|
8 |
59,16599 |
40,75431 |
0,381576 |
0,208923 |
5,912963 |
5,976176 |
|
9 |
56,07245 |
51,23371 |
0,363191 |
0,149877 |
6,924347 |
8,319893 |
|
10 |
46,16116 |
75,611 |
0,268052 |
0,010499 |
10,33103 |
6,864535 |
|
|
|
|
|
|
|
|
|
|
|
|
|
|
|
|
|
|
|
|
|
|
|
|
|
|
|
|
|
|
|
|
|
Analyse de variance: deux facteurs avec répétition d'expérience |
||||||
|
|
|
|
|
|
|
|
|
RAPPORT DÉTAILLÉ |
SOD-C |
GPx-B |
GPx-C |
MDA-B |
MDA-C |
Total |
|
63,7462 |
|
|
|
|
|
|
|
Nombre d'échantillons |
10 |
10 |
10 |
10 |
10 |
50 |
|
Somme |
510,0634 |
9,599856 |
4,835856 |
36,18509 |
40,17974 |
600,8639 |
|
Moyenne |
51,00634 |
0,959986 |
0,483586 |
3,618509 |
4,017974 |
12,01728 |
|
Variance |
66,46867 |
0,045654 |
0,027992 |
1,507012 |
1,049368 |
402,4804 |
|
|
|
|
|
|
|
|
|
55,0725 |
|
|
|
|
|
|
|
Nombre d'échantillons |
10 |
10 |
10 |
10 |
10 |
50 |
|
Somme |
459,6676 |
5,060916 |
3,474874 |
57,60407 |
56,04493 |
581,8524 |
|
Moyenne |
45,96676 |
0,506092 |
0,347487 |
5,760407 |
5,604493 |
11,63705 |
|
Variance |
165,7854 |
0,020557 |
0,030162 |
3,822657 |
2,626797 |
337,9317 |
|
|
|
|
|
|
|
|
|
Total |
|
|
|
|
|
|
|
Nombre d'échantillons |
20 |
20 |
20 |
20 |
20 |
|
|
Somme |
969,731 |
14,66077 |
8,310729 |
93,78917 |
96,22468 |
|
|
Moyenne |
48,48655 |
0,733039 |
0,415536 |
4,689458 |
4,811234 |
|
|
Variance |
116,6986 |
0,085579 |
0,032421 |
3,731877 |
2,403721 |
|
|
|
|
|
|
|
|
|
|
|
|
|
|
|
|
|
|
ANALYSE DE VARIANCE |
|
|
|
|
||
|
Source des variations |
Somme des carrés |
Degré de liberté |
Moyenne des carrés |
F |
Probabilité |
Valeur critique pour F |
|
Échantillon |
3,614379 |
1 |
3,614379 |
0,149735 |
0,699702 |
3,946876 |
|
Colonnes |
33947,71 |
4 |
8486,928 |
351,5941 |
5,02E-54 |
2,472927 |
|
Interaction |
160,0189 |
4 |
40,00471 |
1,657304 |
0,166901 |
2,472927 |
|
A l'intérieur du groupe |
2172,458 |
90 |
24,13843 |
|
|
|
|
|
|
|
|
|
|
|
|
Total |
36283,81 |
99 |
|
|
|
|
|
|
|
|
ctr vs MS+MF |
|
|
|
|
|
|
|
|
|
|
|
|
|
|
|
|
|
GROUP |
SOD-B |
SOD-C |
GPx-B |
GPx-C |
MDA-B |
MDA-C |
|
|
|
1 |
63,74616 |
51,85262 |
1,258829 |
0,657872 |
1,802215 |
3,890363 |
|
|
|
2 |
44,76248 |
57,59533 |
1,125189 |
0,457872 |
1,798677 |
1,802545 |
|
|
|
3 |
53,76248 |
45,81132 |
0,813979 |
0,357872 |
4,034231 |
4,269943 |
|
|
|
4 |
42,8869 |
51,79693 |
0,725189 |
0,721257 |
4,39056 |
4,357405 |
|
|
|
5 |
49,97569 |
70,64803 |
0,894174 |
0,157257 |
4,025063 |
4,627405 |
|
|
|
6 |
38,81047 |
48,57686 |
0,796642 |
0,520876 |
4,650167 |
3,981153 |
|
|
|
7 |
40,74616 |
48,39999 |
0,725189 |
0,385143 |
5,398247 |
5,290618 |
|
|
|
8 |
59,69098 |
41,43127 |
1,179664 |
0,656643 |
3,568593 |
2,794091 |
|
|
|
9 |
47,5936 |
48,35492 |
1,228214 |
0,457872 |
2,385463 |
4,908817 |
|
|
|
10 |
43,5936 |
45,59613 |
0,852788 |
0,463191 |
4,131877 |
4,257405 |
|
|
|
1 |
57,4043 |
54,13869 |
1,071799 |
0,89152 |
2,075024 |
3,934194 |
|
|
|
2 |
48,41443 |
43,99699 |
0,701627 |
0,245932 |
6,430128 |
2,438895 |
|
|
|
3 |
30,42553 |
39,08367 |
0,685831 |
0,088224 |
6,474721 |
5,483333 |
|
|
|
4 |
42,3555 |
45,45747 |
0,694304 |
0,405092 |
4,254798 |
4,890363 |
|
|
|
5 |
44,26849 |
70,64803 |
0,613901 |
0,24449 |
6,224846 |
4,621922 |
|
|
|
6 |
36,57394 |
34,50365 |
0,494304 |
0,308923 |
5,227525 |
4,299097 |
|
|
|
7 |
37,97365 |
33,21414 |
0,408645 |
0,252476 |
7,390394 |
8,348693 |
|
|
|
8 |
45,65923 |
37,10289 |
0,395487 |
0,307347 |
7,561578 |
4,457661 |
|
|
|
9 |
32,53338 |
31,06784 |
0,080521 |
0,073918 |
9,592987 |
6,339093 |
|
|
|
10 |
30,47942 |
36,37993 |
0,264973 |
0,21424 |
3,077398 |
5,575846 |
|
|
|
|
|
|
|
|
|
|
|
|
|
|
|
|
|
|
|
|
|
|
|
|
|
|
|
|
|
|
|
|
|
|
|
|
|
|
|
|
|
|
|
Analyse de variance: deux facteurs avec répétition d'expérience |
|
||||||
|
|
|
|
|
|
|
|
|
|
|
|
RAPPORT DÉTAILLÉ |
SOD-C |
GPx-B |
GPx-C |
MDA-B |
MDA-C |
Total |
|
|
|
63,7462 |
|
|
|
|
|
|
|
|
|
Nombre d'échantillons |
10 |
10 |
10 |
10 |
10 |
50 |
|
|
|
Somme |
510,0634 |
9,599856 |
4,835856 |
36,18509 |
40,17974 |
600,8639 |
|
|
|
Moyenne |
51,00634 |
0,959986 |
0,483586 |
3,618509 |
4,017974 |
12,01728 |
|
|
|
Variance |
66,46867 |
0,045654 |
0,027992 |
1,507012 |
1,049368 |
402,4804 |
|
|
|
|
|
|
|
|
|
|
|
|
|
57,4043 |
|
|
|
|
|
|
|
|
|
Nombre d'échantillons |
10 |
10 |
10 |
10 |
10 |
50 |
|
|
|
Somme |
425,5933 |
5,411392 |
3,032164 |
58,3094 |
50,3891 |
542,7353 |
|
|
|
Moyenne |
42,55933 |
0,541139 |
0,303216 |
5,83094 |
5,03891 |
10,85471 |
|
|
|
Variance |
144,2046 |
0,076272 |
0,052506 |
5,005097 |
2,465839 |
289,5046 |
|
|
|
|
|
|
|
|
|
|
|
|
|
Total |
|
|
|
|
|
|
|
|
|
Nombre d'échantillons |
20 |
20 |
20 |
20 |
20 |
|
|
|
|
Somme |
935,6567 |
15,01125 |
7,868019 |
94,49449 |
90,56884 |
|
|
|
|
Moyenne |
46,78283 |
0,750562 |
0,393401 |
4,724724 |
4,528442 |
|
|
|
|
Variance |
118,5694 |
0,103921 |
0,046692 |
4,372802 |
1,93939 |
|
|
|
|
|
|
|
|
|
|
|
|
|
|
|
|
|
|
|
|
|
|
|
|
ANALYSE DE VARIANCE |
|
|
|
|
|
||
|
|
Source des variations |
Somme des carrés |
Degré de liberté |
Moyenne des carrés |
F |
Probabilité |
Valeur critique pour F |
|
|
|
Échantillon |
33,78934 |
1 |
33,78934 |
1,529601 |
0,21939 |
3,946876 |
|
|
|
Colonnes |
31565,44 |
4 |
7891,359 |
357,2319 |
2,56E-54 |
2,472927 |
|
|
|
Interaction |
353,6962 |
4 |
88,42405 |
4,002845 |
0,00492 |
2,472927 |
|
|
|
A l'intérieur du groupe |
1988,127 |
90 |
22,0903 |
|
|
|
|
|
|
|
|
|
|
|
|
|
|
|
|
Total |
33941,05 |
99 |
|
|
|
|
|
- Judging from some figures and the significance marks, some figures probably used standard deviation and not the standard error of the mean. Please take a second look and confirm that.
No, we have calculated SEM. The standard deviation (SD) measures the amount of variability, or dispersion, from the individual data values to the mean, while the standard error of the mean (SEM) measures how far the sample mean (average) of the data is likely to be from the true population mean, it’s a measure of presicion. We mentioned in the paper the nbval, SEM= standard deviation and dividing it by the square root of the sample size. Sd is calculeted to measure the deviation of a sample, while SEM is a deviation for the true population ‘s mean (the interval in which the true value of the means population could be)
- Some figures are not well aligned.
We have aligned the figures as best as we could.
- Please stress the novel and significant findings your paper has provided in both the introduction and conclusion.
We have changed the introduction in order to make the hypothesis more adequate: “Our hypothesis was that the intensity of cumulative stress comprising of early and adult life stressors would alter intestinal motility, affective and cognitive states in mice more than either of the separate stressors. We also assumed that a disturbed antioxidative balance in brain and in colon as a result of psychological stress would accompany and correlate with behavioral changes, which might help to explain the biochemical mechanisms underlying IBS“ and we added any findings/novelties in the final part of discussions, as follows: “In summary, the present data show that exposure to combined early and adult life stress including an original sequence of predictable and unpredictable stressors results in significant altered intestinal tranzit, anxiety and depression-like behaviors and decreased short term cognitive capacity. Its more significant effect compared to either of single stressors was accompanied by increased oxidative stress in colon and predominantly in brain, which suggests the involvement of a neurologic component in the pathogenesis of IBS. Out of the two separate stressors, maternal separation and multifactorial stress, we found only for the latter a similar pattern with the combined stress group in terms of oxidative stress markers dynamics, suggesting that what is driving the effect is mostly the multifactorial stress exposure. This would not rule out the impact of the MS factor on the antioxidant balance, as some oxidative aggravations were observed in its case as well, but may point to different oxidative pathways or even to some degree of inconsistency in the MS protocol as some studies suggested previously [70]. Heterotypical stress exposure could arguably determine a more immediate central response than early-life stress, but, overall, the combination of the two types of stressors reflected more accurately IBS visceral and affective specific symptoms, advocating for a more precise IBS model in mice.”